# Are AlphaZero-like Agents Robust to Adversarial Perturbations?

**Li-Cheng Lan**[1]    **Huan Zhang**[2]    **Ti-Rong Wu**[3]

**Meng-Yu Tsai**[4]    **I-Chen Wu**[3, 4]    **Cho-Jui Hsieh**[1]

[1]UCLA    [2]CMU    [3]Academia Sinica, Taiwan    [4]NYCU

lclan@cs.ucla.edu   huan@huan-zhang.com   tirongwu@iis.sinica.edu.tw
adam0923686343@gmail.com   icwu@cs.nctu.edu.tw   chohsieh@cs.ucla.edu

## Abstract

The success of AlphaZero (AZ) has demonstrated that neural-network-based Go AIs can surpass human performance by a large margin. However, do these superhuman AZ agents truly learn some general basic knowledge that can be applied to any legal state? In this paper, we first extend the concept of adversarial examples to the game of Go: we generate perturbed states that are "semantically" equivalent to the original state by adding meaningless actions to the game, and an adversarial state is a perturbed state leading to an undoubtedly inferior action that is obvious even for amateur players. However, searching the adversarial state is challenging due to the large, discrete, and non-differentiable search space. To tackle this challenge, we develop the first adversarial attack on Go AIs that can efficiently search for adversarial states by strategically reducing the search space. This method can also be extended to other board games such as NoGo. Experimentally, we show that both Policy-Value neural network (PV-NN) and Monte Carlo tree search (MCTS) can be misled by adding one or two meaningless stones; for example, on 58% of the AlphaGo Zero self-play games, our method can make the widely used KataGo agent with 50 simulations of MCTS plays a losing action by adding two meaningless stones. We additionally evaluated the adversarial examples found by our algorithm with amateur human Go players, and 90% of examples indeed lead the Go agent to play an obviously inferior action. Our code is available at https://PaperCode.cc/GoAttack.

## 1   Introduction

AlphaZero (AZ) [1] like algorithms have achieved state-of-the-art in Go – one of the most challenging games for artificial intelligence. The success of Go AIs like AZ can be contributed to the use of Policy-Value Neural Networks (PV-NN), and Monte Carlo tree search (MCTS) [2, 3]. Silver et al. [4] demonstrated that pure PV-NN can achieve human professional level (Elo 3055) even without any lookahead using MCTS. It has been widely believed that AZ-based agents significantly outperform humans, and even professional Go players can often learn novel strategies from these Go AIs. However, we question if the agents truly learn some basic but general knowledge that can be applied to any legal state.

On the other hand, it is well-known that deep neural networks can be easily fooled by "adversarial examples", which are created by adding small and semantically invariant perturbations to benign inputs [5, 6]. This naturally leads to the following question: are AZ-like Go agents robust to adversarial perturbations? Although adversarial examples have been well-studied in many applications

36th Conference on Neural Information Processing Systems (NeurIPS 2022).

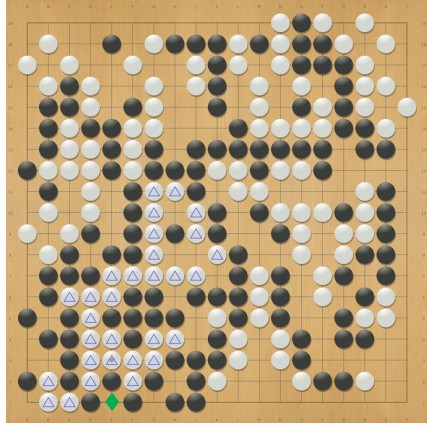
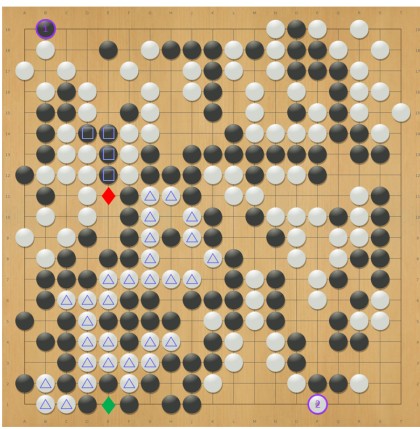

(a) KataGo plays black at E1 ♦
before perturbation.

(b) KataGo plays black at E11 ♦ after adding
two meaningless stones marked as 1 and 2.

Figure 1: Fig. 1b is an adversarial state perturbed from an AlphaGo Zero self-play record (Fig. 1a). After adding two meaningless stones (marked as 1 and 2), a well-trained KataGo agent with 50 MCTS simulations will switch its best action from playing black at ♦ (E1) to ♦ (E11). Even amateur human players can tell that playing at ♦ is wrong since playing black at ♦ can kill all the white stones marked with triangles. However, confused by the adversarial perturbation, KataGo ends up playing ♦ to save the four black stones marked with squares and gives up the opportunity to occupy much more territories by playing at ♦.

of deep neural networks, they have not been thoroughly explored in Go agents because of several challenges. First, the states of Go are discrete. We cannot directly apply gradient-based attacks in this setting, and efficiently searching a large combinatorial space is challenging. Second, Go agents use MCTS to systematically search for the best action, which can be much harder to mislead - in fact, an agent would be able to always find the best action by enumerating all possible outcomes if infinite steps of MCTS simulations were allowed. Third, unlike computer vision and NLP, humans can not serve as the oracle for superhuman Go agents. Hence, even if the target agent does make a mistake, we may not be able to identify it. Because of these challenges, adversarial attack algorithms that have been developed for computer vision [6], NLP [7, 8], and reinforcement learning [9, 10, 11, 12] can hardly be applied to attacking Go agents.

In this paper, we first give the definition of adversarial perturbations for Go. We restrict the adversary to perturb the state (game board) by only playing one or two "meaningless" actions that do not affect the win rate and the game's best action. That is, the perturbed state is "semantically similar" to the original one. An adversarial example is defined as a perturbed state leading to an undoubtedly inferior action from the Go agent that is obvious even for Go beginners. Then, we proposed a novel method to systematically find adversarial states where the AZ agents perform much worse than usual. We carefully designed the constraints on perturbed states during the search so that they are semantically similar to the original states and are also easy enough for human players to verify the correct move. If a Go agent cannot find the correct move for this perturbed state, we say we found an adversarial example. Finally, we test AZ-based agents with thousands of these perturbed states. Surprisingly, we find that agents make trivial mistakes on these adversarial examples, even when MCTS is used.

Fig. 1b shows one of the perturbed states found by our algorithm that AZ agents will fail. B19 and P1 (marked in purple) are the two "meaningless" stones we added as adversarial perturbations. Adding them won't change the turn player's best action and win rate. However, those actions can mislead KataGo [13], a well-known Go agent with 50 MCTS simulations, to "forget" to kill the white stones marked with squares (action A) and instead want to save the black stones marked with triangles (action B), even if doing so will be at a big disadvantage and likely to lose the game. This mistake is very trivial such that even amateur human players can identify it.

The contributions of this paper can be summarized below:

- For the first time, we reveal the vulnerability of both PV-NN and MCTS-based AZ agents. Our proposed attack found adversarial examples (e.g., Fig. 1b) that are semantically equivalent to a "natural" state (from a real game record) while leading to catastrophic behavior of the target agent. The mistakes made by a super-human agent can be identified by Go amateur players.

- Our attack involves an efficient method to speed up the search of adversarial states in the large discrete state space without relying on gradients and is usually more than 100 times faster than brute force search.

- We conduct a comprehensive study on the robustness of four state-of-the-art public AZ agents with six different datasets for the game of Go. By generating adversarial states with two meaningless moves, our attack can consistently achieve over 90% success rates on PV-NNs and achieve over 58% success rates on MCTS agents with 50 simulations. Moreover, our method can also be applied to other games: we found adversarial examples on 50% of data in our NoGo [14] experiment.

## 2    Background and Related Works

**Terminology**    Games like Go can be described as a two-player zero-sum deterministic game [15] and can be defined as a tuple $\langle \mathcal{S}, \mathcal{A}, \mathcal{T}, \mathcal{R} \rangle$, where $\mathcal{S}$ is the state space, $\mathcal{A}$ is the action space, $\mathcal{T} : \mathcal{S} \times \mathcal{A} \mapsto \mathcal{S}$ is the transition function and $\mathcal{R} : \mathcal{S} \mapsto \{1, 0, -1\}$ is the reward. Each game starts from an initial state $s_0 \in \mathcal{S}$ (empty board) at time 0. At the current state $s_t$, the turn-player, defined as $s_t.c$, can play an action $a_t \in \mathcal{A}(s_t)$. In Go, players take turns placing a stone of their color on the board. Therefore, all actions are composed of a color $c$ and a position $p$, indicating placing a $c$ stone on $p$. The only exception is the pass move $a_{\text{pass}}$, which means the turn-player does not place any stone (gives up on his turn). After playing action $a_t$ on state $s_t$, we can get the next state by $s_{t+1} = \mathcal{T}(s_t, a_t)$. The game ends when reaching a terminal state $s_T$, where we can determine who wins the game and each player receives a reward. Since the games are zero-sum, the reward of the opponent player is $-\mathcal{R}(s)$. For the non-terminal states, the reward is $\mathcal{R}(s) = 0$ for any player.

**Policy Value Neural Network (PV-NN) and Policy Value MCTS (PV-MCTS)**    PV-NN is proposed in AlphaGo Zero paper [4]. PV-NN takes a state $s$ as input and outputs a value $v(s)$ and a policy $p(s)$. Value $0 \le v(s) \le 1$ is a scalar from 0 to 1 that indicates the estimated win-rate of current state. For example, $v(s) < 0.5$ means that PV-NN believes the turn-player of state $s$ is losing. Note that if the value output range is $-1 \le v(s) \le 1$, the win-rate can be calculated by $(v(s) + 1)/2$. Policy $p(s)$ is a vector that indicates each action $a$'s probability $0 \le p(a|s) \le 1$. A high probability for an action, e.g., $p(a|s) > 0.9$, indicates that the the PV-NN highly recommends action $a$ at state $s$. Besides PV-NN, AlphaGo Zero [4] also adopted Policy Value MCTS (PV-MCTS) to choose the best action. Unlike MCTS, PV-MCTS uses the value output of PV-NN instead of Monte Carlo simulation to evaluate the selected state values. Moreover, PV-MCTS utilized the policy of PV-NN to narrow down the search space with the PUCT algorithm [16]. The search efficiency of PUCT highly depends on the prior probability $P(s, a^*)$ of best action $a^*$. Therefore, if the root's prior policy is incorrect, the search efficiency will drop dramatically. Take Fig. 1b as an example; even after 50 MCTS simulations, KataGo still hasn't discovered action E1 ♦ is a good action because the prior probability of action E1 ♦ is too small. As more states have been evaluated by PV-NN, PV-MCTS can provide stronger policy $\pi(s)$, value $V(s)$, and action values $Q(s, a)$ of a given root state $s$.

**Adversarial Example**    It has been observed that many neural networks used in computer vision, NLP, and reinforcement learning are vulnerable to adversarial examples [5, 17, 9]. Traditionally, an adversarial example, created by minimally modifying a natural example, is semantically equivalent to the original natural example for humans but can make the target model produce a totally different output. Note that the perturbed state should also look natural [18]. To define an adversarial example for a particular task, one has to define a reasonable perturbation set around original examples and design an algorithm to find an example within the set that leads to incorrect behavior of the model (e.g., misclassification). In computer vision, the perturbation set is usually defined as a small $\ell_p$ norm ball, as small perturbations to images are usually imperceptible to humans. And due to the continuous search space, existing image attacks often rely on gradient-based optimization to find adversarial examples [19, 20, 21, 22]. On the other hand, for **discrete models** like text, there exists multiple definitions for the perturbation set such as synonym substitution [23, 24], edit distance [25], or language model-based scores [26]. For example, if the maximum edit distance is one, given a natural sentence, "This movie had terrible acting." an adversarial example can be "This movie

had awful acting." where we only change one word in the sentence to its synonym. Note that such examples are not humanly imperceptible, and the synonyms are defined by humans. Due to the discrete nature of the text domain, finding an adversarial example leads to a combinatorial search problem, and several attacks have been proposed for this [27, 28, 23]. However, none of these attack methods can be directly applied to attack AZ agents for the following reasons: 1) Unlike the image or NLP domain, the semantically invariant perturbation in Go is difficult to define (see Sec. 3.1). 2) Unlike other applications, the AI's ability is much stronger than humans in Go, so it is non-obvious how to find adversarial examples that can be understood by humans. Although some recent works have studied perturbations to states [12], actions [29], observations [9, 10, 30, 31] or rewards [32] in the reinforcement learning setting, none of them consider discrete input domains like Go or against super-human planning based agent like AZ agents.

**Studies on the weaknesses of Go agents**  Go AI agents have several weaknesses. First, it is well-known that the existence of several patterns such as ladders and long dragons will confuse the agents. This is due to the local nature of convolutional neural networks (CNN). Second, when researchers use Go AI to study Tesuji problems [33], they observe that some of the problems require millions of simulations to find the correct action. Third, some researchers [13] find some states that the agent will ignore the best action (prior probability almost equal to zero), which sometimes refer as "blind spots" of the agents. Our work is different from those works in the following ways. First, our adversarial examples can find bugs at which the model (agent) is used to be good. The mistakes caused by ladders and long dragons are some bugs the model will make in normal states. Second, we find the adversarial examples automatically and systematically. We show that $\geq 58\%$ of the games we selected exist adversarial examples that normal humans can be better than state-of-the-art AIs. By finding "bugs" efficiently, researchers may have enough training data to improve their AIs.

## 3   Method

Given a state $s$, a target agent (e.g., PV-NN or AZ agents), and a much weaker verifier (e.g., humans), our goal is to find an adversarial example $s'$ that satisfies the following conditions:

- $C_1$  The perturbed state $s'$ is very close to the original state $s$ in terms of $\ell_0$ distance.
- $C_2$  The perturbed state $s'$ is semantically equivalent to $s$, and the verifier can verify that.
- $C_3$  The target agent performs correctly on $s$ but wrongly on $s'$, and the verifier can identify that.

Both $C_1$ and $C_2$ define a set of perturbed states, $\mathcal{B}(s)$, for a given natural state $s$. The problem of finding adversarial examples is then equivalent to finding a $s' \in \mathcal{B}(s)$ that satisfies the success criterion ($C_3$). In previous image attacks, $\mathcal{B}(s)$ is a small $\ell_p$ ball [6, 19] and in text attacks $\mathcal{B}(s)$ is usually defined by word substitutions with synonyms [23]. However, in our case, the definition of $\mathcal{B}(s)$ is much more sophisticated and computationally expensive to check.

The existence of a much weaker verifier is to define the adversarial examples we want to find. We hope that the target agent's mistake on the adversarial examples is verifiable to the verifier. Hence, the playing strength of the verifier determines the difficulty of finding adversarial examples. The weaker the verifier is, the harder it is to find an example. In addition, our method finds the $s'$ without the help of verifiers since verifiers, like humans, are hard to include in an automatic process. Therefore, we need to carefully design the $\mathcal{B}(s)$ and the success criterion so that the $s'$ satisfies the conditions automatically. In the following, we will discuss how to define $\mathcal{B}(s)$ and the success criterion in Subsections 3.1 and 3.2, respectively. A search algorithm will then be introduced in Subsection 3.3 to speed up the search.

### 3.1   Defining the Perturbation Set

We define the perturbation set $\mathcal{B}(s)$ of a given state $s$ based on conditions $C_1$ and $C_2$. For $C_1$, we use actions as perturbation and the number of actions we added as distance. In our experiments, we set distance $\ell_0$ as 2. Since each state in Go can be represented by its trajectory (a list of actions), the perturbed state $s'$ can be presented as $a_0, a_1, \ldots, a_{t-1}, b_0, b_1$, where $a_0, \ldots, a_{t-1}$ is the trajectory of $s$ and $b_0, b_1$ are the extra actions. For **1STEP** attack, one of the $b_0, b_1$ has to be the pass action (denoted as $a_{\text{pass}}$). This means we place an additional stone on the board without changing the turn player. On the other hand, for **2STEP** attack, both $b_0, b_1$ are not the pass action, which means we add

one black and one white stone on the board. We do not include states created by replacement like word substitution in text attacks since there are no similar actions in Games like Go.

For condition $C_2$, we aim to maintain the semantic meaning after perturbation. Since there is no ground truth in Go, we resort to an "**examiner**" agent to verify the equivalence of states. The examiner should be able to provide an accurate value $V(s)$, the policy $\pi(s)$, and the action values $Q(s, a)$ for a given state $s$. For example, the examiner we used is the strongest PV-MCTS agent that runs 800 simulations. To distinguish the output of the examiner and the target agent, we use $v(s)$ and $p(a|s)$ as the output of the target agent. With the examiner, we define two states $s, s'$ are semantically equivalent if the two states have the same turn players and similar winrate according to the examiner, which is shown in the following equation.

$$s.c = s'.c \text{ and } |V(s) - V(s')| \le \eta_{\text{eq}}, \tag{1}$$

where $s.c$ denotes the turn player ($c$ for color) of state $s$ and $\eta_{\text{eq}}$ is the threshold used to define that the win rate is close.

Although the examiner is normally better than the target agent, it is still not perfect. The examiner may also make a low-level mistake on $s'$ since the PV-NN it uses is wrong on $s'$. To improve the examiner on $s'$, we provide it with its best action $a_s^*$ on $s$ as a hint. That is, we evaluate the examiner value of $s'$ by forcing the examiner to evaluate the state $\mathcal{T}(s', a_s^*)$. Since $\mathcal{T}(s', a_s^*)$ has a different turn player, so the new $V(s')$ is

$$\max(V(s'), 1 - V(\mathcal{T}(s', a_s^*))). \tag{2}$$

Note that one can provide more hints to the examiner by forcing it to consider more actions or even paths that are reasonable in $s$. This method makes it possible to find adversarial examples even if the target agent is the same as the examiner.

Besides ensuring $s, s'$ to be semantically equivalent, such equivalence should be verifiable to the much weaker verifier like humans. Unlike standard image or text applications where humans are treated as oracles, the AI agents for Go are much more powerful than a human. When the game is too complicated, humans cannot even tell who has the advantage. Therefore, even though both states $s$ and $s'$ are semantically equivalent (Eq. 1) to the examiner, the verifier may not be able to tell.

Fortunately, we observe that humans can identify most of the "meaningless" actions which do not help their turn players to gain any benefits. Here, we define an action as meaningless if its effect on the winrate is equal to playing a pass action. The following is the formal definition:

$$|V(s) - V(F(s, a))| \le \eta_{\text{eq}}, \tag{3}$$

where $F(s, a) : \mathcal{S} \times \mathcal{A} \mapsto \mathcal{S}$ returns the state after playing action $a$ on state $s$ without changing the turn player by playing an additional pass action (Appendix F). Based on this definition, all the actions that satisfy Eq. 1 in the 1STEP attack are already "meaningless" since one of $b_0, b_1$ is a pass action. Therefore, humans can verify that $s$ and $s'$ are semantically equivalent in the 1STEP attack. For 2STEP attack, we require both $b_0, b_1$ to be meaningless (Eq. 3). In this way, humans can verify that $s$ and $s'$ are semantically equivalent by checking that both $b_0$ and $b_1$ are meaningless.

In addition, in the game of Go, we found that most meaningless actions are in one of the player's territories (Appendix E). Also, humans can usually verify such actions faster. Therefore, in the experiments of Go, we further restrict the perturbation action's position $a.p$ within one of the territories $a.p \in \text{Terr}_W \cup \text{Terr}_B$, where $\text{Terr}_W$ and $\text{Terr}_B$ is the territory of each color. The results show that we can find adversarial examples even with this stronger constraint. See Appendix G for the comparison without the territory constraint.

## 3.2 Success Criteria of Adversarial Attack

Next, we define our attack's success criteria ($C_3$). We consider **two types of attacks**: value attacks and policy attacks. For **value attack**, the goal is to change the prediction of the target agent's value network. Therefore, we define the attack to be successful if

$$|v(s) - V(s)| \le \eta_{\text{correct}} \text{ and } |v(s') - V(s')| \ge \eta_{\text{adv}}. \tag{4}$$

The first criterion ensures that the target agent produces the correct value on the original state $s$, and the second criterion ensures that the target value network becomes incorrect after the perturbation. Note that we already enforce $V(s) \approx V(s')$ in Subsection 3.1, so (4) implies that the perturbation will change the target agent's value but not the examiner's value. Constants $\eta_{\text{correct}}$ and $\eta_{\text{adv}}$ are the thresholds to define "correct" and "wrong" in $C_3$. For example, if we want to find an example that PV-NN misclassified the winner of a state, we can set $\eta_{\text{adv}} = 0.5$. We can further increase $\eta_{\text{adv}}$ if we want the target's output to be wrong by a larger margin. In addition, the verifier can easily tell that target is wrong since $s, s'$ are supposed to have the same winrate ($C_2$) but $v(s)$ and $v(s')$ are different.

For **policy attack**, we aim to fool the policy output. However, unlike classification settings, there is more than one best action for a state. Therefore, even when the perturbation can significantly change the output policy, it doesn't mean the new policy is incorrect. We thus have to define the successfulness of the policy attack by checking whether the value (computed by the examiner) will be changed after playing the predicted action. With these in mind, we say the policy attack leads to a "wrong" move if

$$|Q(s, a_s^*) - V(s)| \leq \eta_{\text{correct}} \text{ and } |Q(s', a_{s'}^*) - V(s')| \geq \eta_{\text{adv}} \tag{5}$$

where $a_s^* = \arg\max_a(p(a|s))$ and $a_{s'}^* = \arg\max_a(p(a|s'))$ are the target agent's recommended actions on $s$ and $s'$. The first criterion in (5) ensures that the target agent can predict a proper action for the original state $s$, and the second criterion checks whether the target agent will output a significantly worse action in the perturbed state $s'$. With Eq (5), although the examiner can verify that the target agent plays a losing action on the perturbed state; the verifier may be uncertain that the action $a_{s'}^*$ is truly bad. Luckily, in our qualitative study, by providing $a_s^*$ to humans as a hint, humans can immediately tell that $a_s^*$ is much more important to play than $a_{s'}^*$ in the state $s'$ and certify that the target agent is wrong on most pairs $(s, s')$ we found. Hence, we add two more restrictions: $|Q(s', a_s^*) - V(s')| \leq \eta_{\text{eq}}$, and $|Q(s, a_{s'}^*) - V(s))| \geq \eta_{\text{adv}}$. These restrictions ensure that $a_s^*$ is a good action and $a_{s'}^*$ is a bad action for both $s$ and $s'$, so that the verifier can use them as an anchor.

### 3.3 An Efficient Attack Algorithm

Like other adversarial attacks on reinforcement learning [9, 10], we apply our attack to perturb a state in a given game since it only takes one mistake for an agent to lose a game. Formally, given a game $G = \{s_0, s_1, \ldots, s_T\}$, if we can find one pair of $(s_i, s_i')$, where $s_i'$ is an adversarial example of $s_i$, then we have successfully attacked the agent on that game. We will first apply the 1STEP attack to a game, and if it fails, we then conduct the 2STEP attack.

Since the search space is countable, a naive way to find an adversarial example is to conduct a brute-force search to check all states in the search space. However, this will lead to very high complexity, and the search cannot be finished in practice. Taking the 2STEP attack as an example, let $T \approx 300$ denote the game length, $N \approx 150$ denote the average number of actions for a state, and $M \geq 800$ denote the simulation count of the examiner. Then, we need $O(TNM)$ time to obtain the search space for

---

**Algorithm 1** Two-Step Value Attack

1: **Input:** a game $s_1, s_2, \ldots, s_T$, target agent $t$, examiner $e$
2: **for** $i = T$ **to** 0 **do**
3:     **if** $|e.V(s_i) - t.v(s_i)| > \eta_{\text{correct}}$ **then**
4:         **continue**
5:     cands = getMeaninglessActions $(e, s_i)$
6:     **for** $b_0$ in cands[0] **do**
7:         **for** $b_1$ in cands[1] **do**
8:             $s' = \mathcal{T}(\mathcal{T}(s_i, b_0), b_1)$
9:             **if** $|t.v(s') - e.V(s_i)| \geq |\eta_{\text{adv}} - \eta_{\text{eq}}|$ **then**
10:                 **if** $|t.v(s') - e.V(s')| \geq \eta_{\text{adv}}$ **and** $|e.V(s) - e.V(s')| \leq \eta_{\text{eq}}$ **then**
11:                     **return** $s'$
12: **return NULL**

---

all $\mathcal{B}(s_i), i \in \{0, 1, \ldots, T\}$ by checking which actions are meaningless. Furthermore, assume that there are averagely $\bar{N}$ *meaningless moves* for each state.

Then, we need $O(T\bar{N}^2M)$ to check the success criteria of each perturbation. Note that in both stages, the running time of the examiner will dominate, as we need to run the examiner with much more MCTS steps. Hence, we propose the following two approaches to reduce the search complexity.

Table 1: Our attack results on 4 Go agents, averaged over all datasets. EXECUTED NUM shows the average number of target and examiner calls for the attack, and SPEEDUP indicates the speedup of the proposed method over the brute-force search. High attack success rates are observed in all settings.

| | AGENT | SUCCESS RATE | | EXECUTED NUM | | SPEEDUP |
|---|---|---|---|---|---|---|
| | | 1STEP | 2STEP | TARGET | EXAMINER | |
| VALUE $\eta_{adv}=0.5$ | KATAGO | 0.99 | 1.00 | 6152 | 68 | 80 |
| | LEELA | 0.90 | 0.98 | 37533 | 182 | 163 |
| | ELF | 0.92 | 1.00 | 11108 | 108 | 90 |
| | CGI | 1.00 | 1.00 | 40 | 2 | 14 |
| VALUE $\eta_{adv}=0.7$ | KATAGO | 0.86 | 0.94 | 44638 | 299 | 125 |
| | LEELA | 0.70 | 0.84 | 96134 | 518 | 150 |
| | ELF | 0.65 | 0.87 | 42503 | 398 | 94 |
| | CGI | 1.00 | 1.00 | 45 | 2 | 15 |
| POLICY $\eta_{adv}=0.5$ | KATAGO | 0.70 | 0.92 | 128228 | 666 | 155 |
| | LEELA | 0.92 | 0.97 | 17218 | 118 | 123 |
| | ELF | 0.93 | 0.97 | 16387 | 113 | 122 |
| | CGI | 0.87 | 0.96 | 16495 | 103 | 133 |
| POLICY $\eta_{adv}=0.7$ | KATAGO | 0.55 | 0.82 | 231895 | 707 | 232 |
| | LEELA | 0.75 | 0.93 | 36438 | 150 | 186 |
| | ELF | 0.77 | 0.88 | 31258 | 159 | 157 |
| | CGI | 0.75 | 0.89 | 24915 | 139 | 146 |

Table 2: Our attack results on 6 datasets, averaged over all Go agents. EXECUTED NUM shows the average number of target and examiner calls for the attack. The first five datasets are described in experiment settings. FOX is the dataset of non-professional players.

| | GAMES | SUCCESS RATE | | EXECUTED NUM | |
|---|---|---|---|---|---|
| | | 1STEP | 2STEP | TARGET | EXAMINER |
| VALUE $\eta_{adv}=0.7$ | ZZ | 0.78 | 0.89 | 14691 | 316 |
| | ZM | 0.90 | 0.95 | 12977 | 197 |
| | MH | 0.60 | 0.81 | 173844 | 663 |
| | LG | 0.85 | 0.95 | 12650 | 161 |
| | ATV | 0.88 | 0.95 | 14988 | 185 |
| | FOX | 0.83 | 0.94 | 53792 | 213 |
| POLICY $\eta_{adv}=0.7$ | ZZ | 0.57 | 0.82 | 33314 | 502 |
| | ZM | 0.84 | 0.94 | 13374 | 178 |
| | MH | 0.51 | 0.80 | 227202 | 258 |
| | LG | 0.84 | 0.96 | 52055 | 274 |
| | ATV | 0.75 | 0.88 | 79688 | 232 |
| | FOX | 0.75 | 0.87 | 141120 | 225 |

First, we reduce the time of finding meaningless actions by the following observation. If an action $a$ is a meaningful action (not meaningless) of state $s_t$, then it is likely that $a$ is also a meaningful action of state $s_{t-1}$. Intuitively, if an action $a$ is a meaningful action at $s_t$, it means that $a$ can occupy some extra territory that is not occupied at $s_t$. Hence, if a position is not occupied in $s_t$, it is likely that the position is not occupied in $s_{t-1}$ and can be occupied by action $a$ too. So action $a$ is also a meaningful action for $s_{t-1}$. We formally prove this property under some assumptions in Appendix A. Based on this observation, we run the search in a backward manner from the final state $s_T$ to the initial state $s_1$. Once an action is identified as meaningful at state $s_t$, we do not need to check it again on any state $\{s_i : i < t\}$. This will significantly save the computational time to enumerate $\mathcal{B}(s)$. The details of getting meaningless actions are shown in Appendix C.

Second, for the part of checking the attack success criterion, we derive a bound to filter out unsuccessful perturbations quickly. Taking the value attack as an example, checking (1) and (4) requires running the examiner on the perturbed state $s'$, which is the bottleneck of the algorithm. To reduce this cost, we show that the condition of (1) and (4) implies

$$
\begin{aligned}
|v(s') - V(s)| &= |(v(s') - V(s')) - (V(s) - V(s'))| \\
&\geq |v(s') - V(s')| - |V(s) - V(s')| \\
&\geq \eta_{\mathrm{adv}} - \eta_{\mathrm{eq}}.
\end{aligned}
\tag{6}
$$

Since (6) only requires running examiner on the original state $s$ instead of $s'$, we can check (6) first before checking (1) and (4). We observe this step can filter out more than 99% of the $s'$.

The overall attack algorithm is presented in Algorithm 1. The input includes a game $s_1, s_2, \ldots, s_T$, a target agent $t$, and an examiner agent $e$. As mentioned before, the search is conducted in a backward manner from $s_T$ back to $s_1$ (line 2). For each $s_i$, we first check whether $s_i$ is too hard for the target (line 3). If so, we will skip this state (line 4). We then compute all the meaningless actions of state $s_i$ using the examiner with the efficient implementation mentioned above and store the meaningless actions separately in cands$[0]$ and cands$[1]$ according to their color (line 5) as the candidates of perturbation. We then check whether each 2STEP perturbation will lead to a successful attack (lines 9-11). Note that line 9 is based on (6), and the examiner does not need to compute $e.V(s_i)$ again since it has evaluated it on line 3. For policy attack, we can skip all the states that are losing $V(s) < \eta_{\mathrm{adv}}$ since for those states, there is no correct answer to attack. (More details in Appendix B).

## 4 Experiments

**Experiment Settings** We evaluate our method on 19x19 Go. The four open-source programs we used are KataGo [13], Leela Zero [34], ELF OpenGo [35], and CGI [36]. The strengths of these

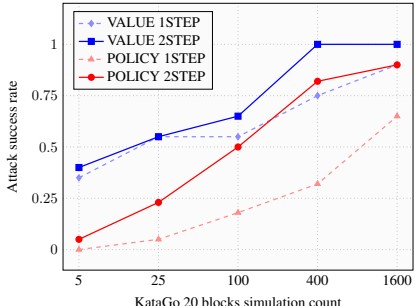

Figure 2: The attack success rate of $\eta_{\text{adv}} = 0.7$ on datasets that are generated by KataGo 40 blocks with 800 simulations vs KataGo 20 blocks with different numbers of simulations.

Table 3: KataGo with different simulations on ZZ dataset. EXECUTED NUM shows the average number of target and examiner calls for the attack.

| | | SUCCESS RATE | | EXECUTED NUM | |
| --- | --- | --- | --- | --- | --- |
| | SIM | 1STEP | 2STEP | TARGET | EXAMINER |
| | 1 | 1.00 | 1.00 | 1016 | 41 |
| | 5 | 0.89 | 0.95 | 4454 | 2322 |
| VALUE | 10 | 0.89 | 0.95 | 6501 | 3147 |
| $\eta_{adv} = 0.5$ | 25 | 0.84 | 0.89 | 18516 | 13903 |
| | 50 | 0.53 | 0.58 | 76426 | 42974 |
| | 1 | 0.84 | 1.00 | 18347 | 869 |
| | 5 | 1.00 | 1.00 | 3309 | 1318 |
| POLICY | 10 | 0.95 | 1.00 | 6666 | 2662 |
| $\eta_{adv} = 0.5$ | 25 | 0.79 | 1.00 | 12466 | 5487 |
| | 50 | 0.21 | 0.68 | 89512 | 41702 |

AI agents are KataGo $\gg$ Leela > ELF = CGI, where KataGo has more than 99% winrate against Leela. We use KataGo (40 blocks) with 800 simulations as our examiner. For the thresholds, we set $\eta_{\text{eq}} = 0.1$, $\eta_{\text{correct}} = 0.15$, since after testing several different $\eta_{\text{eq}}$ and $\eta_{\text{correct}}$ values, this pair leads to more human-understandable results. For the datasets, we selected 99 games from five different sources, which are AlphaGo Zero 40 blocks training self-play record (ZZ), AlphaGo Zero vs AlphaGo Master (ZM), AlphaGo Master vs Human champions (MH), the final games of LG Cup World Go Championship (2001-2020) (LG), and the final games of Asian TV Cup (2001-2020) (ATV). Note that the thinking time for ATV Cup is much shorter than LG Cup, so we expect them to reflect human games with different strengths. All the datasets have 20 games, except ZZ has 19 games since the first game is played by two random agents.

**Results on Different PV-NNs** We first evaluate the robustness of the four agents' PV-NNs since PV-MCTS is much slower. The results are shown in Table 1, where we attack each agent's PV-NN with both 1STEP and 2STEP attacks with various $\eta_{\text{adv}}$. We also present the average number of evaluations required for the target agent and the examiner agent to show the speedup. The first two groups in Table 1 demonstrate the robustness of these agents against the value attack. Group one ($\eta_{\text{adv}} = 0.5$) shows that even the 1STEP attack can achieve above 90% success rate on all agents and mostly achieve 100% on the 2STEP attack. For those that have the same attack success rate, we can still compare the attack difficulties by the number of target evaluations. For example, although the 2STEP success rates of KataGo, ELF, and CGI are all 100%, the number of states that they have visited is totally different. For CGI, we are able to find an adversarial example after visiting 40 states, while ELF requires visiting 11,108 states. Hence, we conclude that Leela > ELF > KataGo > CGI in terms of their robustness against value attack when $\eta_{\text{adv}} = 0.5$ and $\eta_{\text{adv}} = 0.7$. Interestingly, this ranking does not match the playing strength of each agent. The third and fourth groups of Table 1 show the results of the policy attacks. In general, we observe that it is harder to attack the policy than the value. Interestingly, the ranking of four agents in terms of their policy's robustness is KataGo $\gg$ Leela > ELF $\approx$ CGI, which aligns with the playing strengths of those agents.

In Table 1, we also demonstrate our algorithm's speedup in the SPEEDUP column compared to the brute force algorithm. In the brute force algorithm, the examiner must evaluate all the states the target agent has evaluated. Hence, the speed up is almost equal to $(n_{\text{target}} \times n_{\text{MCTS}})/(n_{\text{target}} + n_{\text{examiner}} \times n_{\text{MCTS}})$, where $n_{\text{target}}$ and $n_{\text{examiner}}$ are the numbers of states that the target and the examiner have executed. $n_{\text{MCTS}}$ is the number of simulations that the examiner use for the PV-MCTS. The results show that the proposed efficient search method is usually more than a hundred times faster than a brute-force search, especially for harder problems that require the 2STEP attacks to succeed.

**Results on Different Datasets** We also study the robustness of all agents on different types of game records. We consider the 5 datasets used in the previous subsection, plus an additional FOX [1] dataset, to represent amateur players and see if the games played by weaker players are harder to attack

---

[1] https://www.foxwq.com/

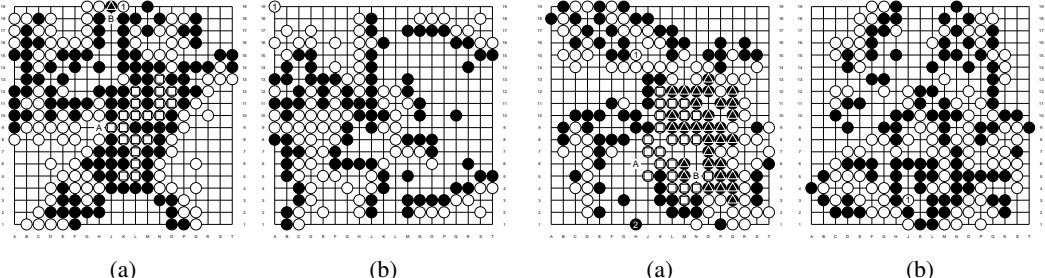

|        (a)         |        (b)         |        (a)         |        (b)         |

Figure 3: Adversarial example of policy attack (a) and value attack (b) on FOX. Both examples are 1STEP attacks with turn player white. Their perturbation actions are marked as 1. For (a), the agent plays B instead of A. For (b), the agent predicts a very different winrate (> 0.9) on the original state and the perturbed state.

Figure 4: Adversarial examples generated by policy attack (a) and value attack (b) that cannot be easily verified by humans. For (a), it is hard for humans to verify that playing white on B is a bad action even if A is provided. For (b), perturbation action marked as 1 does help white get some benefits, but not enough to change the winrate. Hence, it is meaningless. However, humans can not verify it unless they thoroughly calculate both sides' territories.

since those games are easier for humans to understand. The results in Table 2 show that PV-NNs are vulnerable to all levels of games, regardless of AI agents, professional players, or non-professional players. We observe that the AlphaGo Master vs. Human champions (MH) dataset has the lowest success rate on both policy and value attacks compared to other datasets. Since AlphaGo Master is much stronger than human champions ($60:0$), we hypothesize that "when one player is much stronger than the other, the winner of the game may be too obvious and easy for agents to judge." To support this hypothesis, we generate games of KataGo 40 blocks (K40) with 800 simulations vs. KataGo 20 blocks (K20) with different simulations to simulate games played by players with different strengths. Note that K20 is already much weaker than K40. Fig. 2 shows the results of attacking KataGo's PV-NN with those games. We can see that when the games have larger gaps, both the policy and value output of KataGo's PV-NN will be harder to attack. This suggests that the playing strength between game players will likely affect the difficulty of finding adversarial examples.

**Robustness of PV-MCTS**   In this paragraph, we investigate the robustness of PV-MCTS with different simulations. Since PV-MCTS is much slower than PV-NN, we only test KataGo on the AlphaGo Zero self-play (ZZ) dataset with $\eta_{adv} = 0.5$. The results are shown in Table 3. The first column is the simulation count of KataGo. When the simulation equals one, it is the same as using PV-NN directly. The first group shows the results of the value attack. Even with 25 simulations, the 2STEP success rate is still 89%. For group two, we observe that the policy of small simulations is even less robust than PV-NN's policy. For example, when the simulations of KataGo is 10, the 1STEP success rate is 95%, while the policy of PV-NN is 84%. Therefore, we conclude that with a small number of MCTS simulations, the agent will not be able to recover from the bad PV-NN outputs and will still be fooled by adversarial examples. However, when using more MCTS simulations, the attack success rates will still drop since the agent has more chances to discover correct actions.

**Quality of adversarial examples**   Although the AZ agents do make mistakes on the adversarial examples we found, we still need to make sure that those mistakes are so low-level that even humans can verify them. Hence, we randomly selected 100 examples from all the experiments and conducted the following two human studies to see how many percent of mistakes are human verifiable mistakes. Fig. 3 4 show four examples that we samples, and more can be found in Appendix H. In the first study, we examine whether humans can verify that perturbed state $s'$ is semantically equal to the original state $s$. For each adversarial example, we present both the original board and the additional stones to three amateur human players (with level 2K, 3D, 5D) who served as verifiers. For 90% of those perturbations, they can certify that the perturbations are meaningless with a short thinking time. Fig. 4 (right) shows one of the fail examples, where the meaningless action J3 seems meaningful to

humans since it can capture the black stone J2. However, with a longer thinking time and discussion with each other, verifiers can understand that the rest 10% of actions are meaningless.

In the second study, we examine whether the "wrong" actions resulting from our policy attack can be verified by humans. Similar to the previous experiment, we present the original board, original action, perturbed board, and the action after perturbation to the examiners. When the target agent is a PV-NN, humans can identify 100% that the recommended actions after perturbation will change the result from winning to losing. However, for the adversarial examples where the target is PV-MCTS agents, only 70% of the recommended actions after perturbation can be identified as wrong by humans. The main reason is that as PV-MCTS agents are able to lookahead, they tend to avoid actions that are clearly wrong, so the errors become more subtle to humans. Fig. 4 (left) shows one of the states, where the white should play at A to save the stones marked with squares, but with two meaningless moves added the PV-MCTS agent with 100 simulations will play at B. Although after playing B all the stones marked with squares will be dead, it is hard for humans to verify the outcome of this play.

**Agents are sensitive to the ordering of actions in the trajectory**    Finally, after viewing those adversarial examples, we find that the policy attack succeeds often because the target agents are too reliant on the information of the last action. For example, if KataGo knows the last action of Fig. 1b besides the actions we add is white playing E3, even PV-NN knows that it needs to play E1. Based on this observation, we can improve the robustness of PV-NNs using the following method. Given a state $s$ with trajectory $a_0, a_1, \ldots, a_t$, we define an augmented state $\bar{s}$ as $a_0, a_1, a_2, \ldots, a_{t-1}, a_t, a_{t-2}, a_{t-3}$. If $\bar{s}$ is a legal state, we can define a robust policy $p_r(a|s) = (p(a|s) + p(a|\bar{s}))/2$. With this method, PV-NNs can evaluate the state $s$ with different actions being the last. We evaluated this method on the same setting as Table 3 (attacking KataGo's PV-NN with ZZ dataset and $\eta_{\text{adv}} = 0.5$). The success rate of 1STEP and 2STEP policy attacks dropped to 58% and 89%, even better than the original PV-NN and PV-MCTS with 25 simulations.

**Experiments on the game of NoGo**    Our "action perturbation" method can also be used to find severe bugs automatically on other games by removing the territory constraint. We use the following 9x9 NoGo [14] experiments to demonstrate it. We use an AZ agent [37] with 800 simulations as the examiner and use its PV-NN as the target agent. Since NoGo is a game without any human experts, we use a traditional MCTS agent[38] with 1000 simulations as the weaker verifier. Note that the target agent has a 100% winrate against the verifier in 1000 games. Intuitively, the target network will not make a critical mistake that the verifier will not make. However, on 20 self-play games of the target network (the average search space is about 10000), 50% of the games exist adversarial policy examples with $\eta_{adv} = 0.5$ that the verifier can verify. Moreover, our proposed efficient search method based on Eq. 6 is 307x faster than the brute force search.

## 5  Conclusion and Future Works

In this paper, we first properly define the perturbation set $\mathcal{B}(s)$ and the success criteria with the help of an examiner, which become even more reliable after giving it some hints. The adversarial examples we find are understandable to a much weaker verifier like amateur human players. We also proposed a new efficient search algorithm by reducing the usage of the examiner. Our experiments found that even the strongest AZ agent with a small number of simulations is vulnerable to our adversarial attack. We hope our work can raise the attention that even for AI agents that surpass humans by a large margin, they still can easily make simple mistakes that humans will not. We also hope that the examples we found can be used to improve AI agents. For example, for Go, we have shown that the agents are sensitive to the ordering of actions in the trajectory.

A limitation of this paper is that we are only able to identify adversarial states but haven't been able to guide the AZ agents to those states systematically. Potentially, this can be done by training an agent to play against a particular target agent and incorporating some blind spots found by our attacks. Although [39] try this after our publication, their agents cannot beat KataGo under realistic judgments, so this is still an interesting open problem to pursue in the future.

Finally, we hope our work can serve as a generalization benchmark for Go AI. If a Go AI is generalized enough, it should be able to provide a decent action on any state in Go, just like humans. One of the solutions is to use the numerous examples we found to train a better agent.

## Acknowledgments and Disclosure of Funding

This work is supported in part by NSF under IIS-2008173, IIS-2048280, and by Army Research Laboratory under W911NF-20-2-0158.

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
