# A  Proof of Meaningful Action

In this section, our goal is to prove that a meaningful action $b_0$ of the state $s_t$ will also be the meaningful action of the state $s_{t-1}$ under following two assumptions.

a$_1$  Changing the order of the trajectory $a_0, a_1, \ldots, a_{t-1}$ of state $s_t$ will not change the state $s_t$.

a$_3$  The action $a_{t-1}$ is the best action of both $s_{t-1}$ and $s_{t-1}'$

a$_2$  Adding an extra action to the board will not reduce the winrate of the action's color.

Both assumptions are true in the most states on Go and NoGo. For convenience, we assume the turn color is of $s_t$ is black and $b_0$ is one of its meaningful action. We also define $V_B(s)$ as the state value of black for state $s$. Now our goal is to prove that $b_0$ is still meaningful action to $s_{t-1}$.

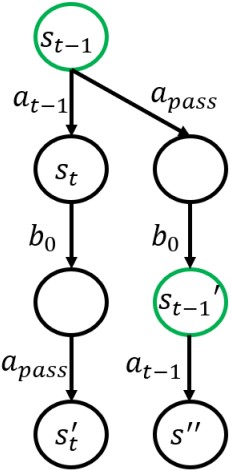

Figure 5: Proof illustration

Since adding an extra action will only benefit to the action's player and the turn player of $b_0$ is black. According to Fig. 5 We just need to prove that

$$\text{Given } V_B(s_t) < V_B(s_t'), \text{ prove that } V_B(s_{t-1}) < V_B(s_{t-1}'). \tag{7}$$

First since $a_{t-1}$ is the best action of $s_{t-1}$, we have $V_B(s_t) = V_B(s_{t-1})$. Also, since $a_{t-1}$ is the best action of $s_{t-1}'$, we have $V_B(s_{t-1}') = V_B(s'')$.

Next, due to the a$_1$ assumption, we have $s_t' = s''$.

Finally, we have $V_B(s_{t-1}') = V_B(s'') = V_B(s_t') > V_B(s_t) = V_B(s_{t-1})$

## B  Algorithm of the 2STEP Policy Attack

The algorithm is shown in algorithm 2.

---
**Algorithm 2** Two-Step Policy Attack

---
1: **Input:** a seq states $s_i$, target agent $t$, examiner $e$
2: **for** $i = T$ **to** $0$ **do**
3:    $a_s = \arg\max_a(t.p(a|s))$
4:    **if** $|e.Q(s_i, a_s) - e.V(s_i)| > \eta_{\text{correct}}$ **or** $e.V(s_i) < \eta_{\text{adv}}$ **then**
5:       **continue**
6:    $actions = \texttt{getMeaninglessActions}(e, s_i)$
7:    **for** $b_0$ in $actions[0]$ **do**
8:       **for** $b_1$ in $actions[1]$ **do**
9:          $s^{'} = \mathcal{T}(\mathcal{T}(s_i, b_0), b_1)$
10:          $a_{s'} = \arg\max_a(t.p(a|s^{'}))$
11:          **if** $|e.Q(s_i, a_{s'}) - e.V(s_i)| \geq \eta_{\text{adv}}$ **then**
12:             **if** $|e.Q(s^{'}, a_{s'}) - e.V(s^{'})| \geq \eta_{\text{adv}}$ **and**
                 $|e.V(s) - e.V(s^{'})| \leq \eta_{\text{eq}}$ **and**
                 $|e.Q(s^{'}, a_s) - e.V(s^{'})| \leq \eta_{\text{correct}}$ **then**
13:                **return** $s^{'}$
14: **return NULL**

---

## C  Algorithm of Getting Meaningless Action

The algorithm is shown in algorithm 3.

---
**Algorithm 3** Get Meaningless Actions

---
1: **Member variable** a set of actions that are meaningful $meaningful\_actions$
2: **Input:** a state $s_i$, target agent $t$, examiner $e$
3: $ret = [[], []]$
4: $terr = e.\text{get\_territory}(s_i)$
5: $a_s = e.\text{get\_best\_action}(s_i)$
6: $V_{s'} = e.\text{get\_value}(\mathcal{T}(s_i, a_s))$
7: **for** $a$ **in** $\mathcal{A}(s_i) \bigcup \mathcal{A}(\mathcal{T}(s_i, a_{pass}))$ **do**
8:    **if** $-0.8 \leq terr[a.p] \leq 0.8$ **then**
9:       **continue**
10:    **if** $a.c == s_i.c$ **then**
11:       $s^{''} = \mathcal{T}(\mathcal{T}(\mathcal{T}(s, a), a_{pass}), a_s)$
12:    **else**
13:       $s^{''} = \mathcal{T}(\mathcal{T}(\mathcal{T}(s, a_{pass}), a), a_s)$
14:    $v_{s''} = e.\text{get\_quick\_value}(s^{''})$
15:    **if** $a$ **in** $meaningful\_actions$ **and** $\text{isEq}(v_{s''}, V_{s'})$ **then**
16:       **if** $\text{isEq}(e.\text{get\_value}(s^{''}), V_{s'})$ **then**
17:          $meaningful\_actions.\text{remove}(a)$
18:    **else if** $a$ **not in** $meaningful\_actions$ **and not** $\text{isEq}(v_{s''}, V_{s'})$ **then**
19:       **if not** $\text{isEq}(e.\text{get\_value}(s^{''}), V_{s'})$ **then**
20:          $meaningful\_actions.\text{add}(a)$
21:    **if** $a$ **not in** $meaningful\_actions$ **then**
22:       **if** $a.c == s_i.c$ **then**
23:          $ret[0].\text{add}(a)$
24:       **else**
25:          $ret[1].\text{add}(a)$
26: **return** $ret$

---

## D   The Input of PV-NN

The inputs of PV-NN are not independent. Take AlphaGo Zero as an example, given a state $s_t$ at step $t$, it will generate 17 feature planes as the input of PV-NN. Each feature plane is a $19 \times 19$ binary 2D array that includes the information the latest eight states. For example, there are eight feature planes $\{X_t, X_{t-1}, \ldots, X_{t-7}\}$ indicates the presence of the current player's stones of $s_t, s_{t-1}, \ldots, s_{t-7}$. If position $p = (i, j)$ has current player's stone at time $t$, then $X_t[i][j] = 1$ else $X_t[i][j] = 0$. Since each player can only place on stone at a time, $X_i$ will mostly be the same. Hence, the legal inputs feature are not independent. Additionally, most AIs have more complex feature planes as input, including the domain knowledge of Go. For example, a common feature is "liberty," which means how many additional enemy stones are needed to capture the position. This kind of input is also not independent to other feature planes.

## E   Territory

Starting from an empty board, one player places a stone on a vacant part of the board to surround more territory or defend our territory from being "captured" by the opponent's stones. It is critical to select actions that can gain more territory. Normally, once a position belongs to a color, it is hard to change it. Hence, putting a stone in any color's territory is usually wasting a turn.

Since territory is so important in Go, many agents' PV-NNs ([13]) has an additional output to predict the territory. Given a state $s$, the territory output $terr(s)$ is a vector of scalar. Each element of the vector $-1 \leq terr(s)[i] \leq 1$ shows how a position $i$ is likely to belong to. For example, $terr(s)[i] <= -0.8$ means that position $i$ is likely belongs to the color white and $terr(s)[i] >= 0.8$ means position $i$ is likely belongs to the color black. In our experiment, we define the positions of meaningless actions should be one of the players' territory. That is, we will not consider an action $a$ as meaningless action if its position $p$ is not one of the player's territory $-0.8 \leq terr(s)[i] \leq 0.8$.

## F   Formal Definition of the Skip Function

Function $F(s, a) : \mathcal{S} \times \mathcal{A} \mapsto \mathcal{S}$ will play the action $a$ on state $s$ without changing the turn player by skipping the opponent's turn. Since action's color $a.c$ might not be the turn color of $s$, in that case, we need to play action pass $a_{pass}$ first. Finally, $F$ is formulated as follow:

$$F = \begin{cases} \mathcal{T}(\mathcal{T}(s, a), a_{\text{pass}}) & s.c = a.c \\ \mathcal{T}(\mathcal{T}(s, a_{\text{pass}}), a) & s.c \neq a.c \end{cases}$$

where $s.c, a.c$ are the color of the state and the action, $a_{\text{pass}}$ is the pass action.

# G Go Experiments without Territory

In this section, we attack KataGo's PV-NN without using the territory constraint. The results are shown in Table 4 and can be compared with Table 5 which is the normal setting with territory. Without the territory constraint, it is easier for our method to attack the target model. For example, the 2STEP success rate on policy attack become 100% after removing the territory constraint. Another observation is that without the territory constraint, it is more likely to call the examiner. This might because the meaningless actions under territory constraint are more stable. Hence, if $v(s')$ is different form $V(s)$, it is more likely that it is and adversarial example, instead of $s'$ is semantically different form $s$.

Table 4: Attack KataGo's PV-NN without territory constraint.

|  |  | SUCCESS RATE | | EXECUTED NUM | |
| --- | --- | --- | --- | --- | --- |
|  | GAMES | 1STEP | 2STEP | TARGET | EXAMINER |
| VALUE $\eta_{adv} = 0.7$ | ZZ | 0.89 | 0.95 | 61164 | 797 |
|  | ZM | 0.90 | 1.00 | 57258 | 2242 |
|  | LG | 0.95 | 0.95 | 60770 | 835 |
|  | ATV | 1.00 | 1.00 | 5447 | 83 |
| POLICY $\eta_{adv} = 0.7$ | ZZ | 0.80 | 1.00 | 58895 | 2716 |
|  | ZM | 0.95 | 1.00 | 16748 | 560 |
|  | LG | 0.60 | 1.00 | 199201 | 4832 |
|  | ATV | 0.75 | 1.00 | 258985 | 5300 |

Table 5: Attack KataGo's PV-NN with territory constraint.

|  |  | SUCCESS RATE | | EXECUTED NUM | |
| --- | --- | --- | --- | --- | --- |
|  | GAMES | 1STEP | 2STEP | TARGET | EXAMINER |
| VALUE $\eta_{adv} = 0.7$ | ZZ | 0.84 | 0.95 | 24044 | 476 |
|  | ZM | 0.85 | 1.00 | 24904 | 347 |
|  | LG | 0.9 | 0.95 | 13405 | 194 |
|  | ATV | 0.95 | 1.00 | 9215 | 138 |
| POLICY $\eta_{adv} = 0.7$ | ZZ | 0.68 | 0.89 | 98739 | 1077 |
|  | ZM | 0.85 | 1.00 | 26028 | 303 |
|  | LG | 0.45 | 0.80 | 295151 | 686 |
|  | ATV | 0.5 | 0.85 | 189696 | 870 |

# H Visualized Examples

Fig. 6 provides some examples we found on different agents and different types of games. The perturbation actions are marked as 1 and 2. If it is a policy attack, the best action of both $s$ and $s'$ is mark as $A$ and the bad action that the target model want to play is marked as $B$. The subcaption of each examples first shows the which dataset it is form. For example, "FOX_12k" means that the game from dataset FOX and was generate from level 12k player. For another example, "ZZ_8" means the number 8 game of AlphaGo Zero self-play. Second, the subcaption describe the target model. If it is just PV-NN, we will present it using the first letter of the agent. For example, 'K' for KataGo, 'C' for CGI. On the other hand, if the target model is MCTS, we will add a 'M' and simulation count after the first letter of the model. For example, "KM25" means KataGo MCTS agent with 25 simulation. Finally, the third component of the subcaption show the type of attack and the threshold $\eta_{adv}$. For example, "P0.7" means policy attack and $\eta_{adv} = 0.7$.

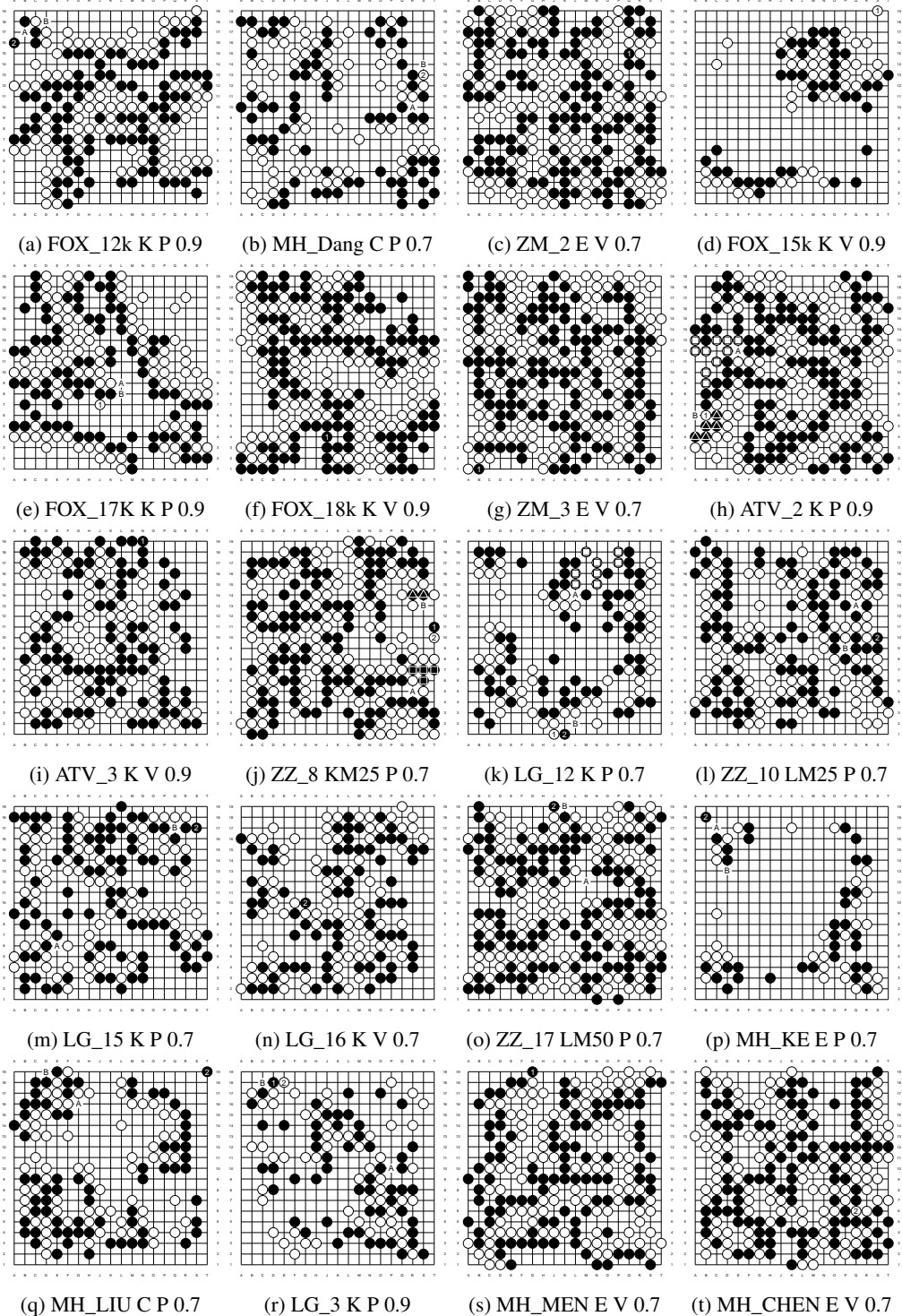

Figure 6: Each subfigure present an adversarial example. The subcaption provides the name of dataset, the program to be attacked (K = KataGo, L = Leela, E = Elf, C=CGI, KMXX = KataGo MCTS with XX simulations), and the policy attack (P) or value attack (V) concatenating with $\eta_{\text{adv}}$, separating by space. For example, in (a), "FOX_12k K P 0.9" represents a policy attack with $\eta_{\text{adv}} = 0.9$ on KataGo, where the game record is chosen from FOX dataset, and in (j) "ZZ_8 KM25 P0.7" means the attacking policy with $\eta_{\text{adv}} = 0.7$ **KataGo MCTS** with **25** simulation on **ZZ** dataset.