# OpenReview forum: "Are AlphaZero-like Agents Robust to Adversarial Perturbations?"
_NeurIPS.cc/2022/Conference — NeurIPS 2022 Accept_

### Official Review · Reviewer_nfAu · 2022-06-30

**Rating:** 3
**Confidence:** 5
**Soundness:** 3 good
**Presentation:** 3 good
**Contribution:** 2 fair

**Summary:**

The author proposes the first adversarial attack on AlphaZero, including value attack and policy attack, to fool the target agent’s value and the policy output respectively. The paper also develops an efficient trick and a bound rule to reduce the time complexity. By adding one or two meaningless stones, agents on different datasets are vulnerable to those “adversarial examples”.

**Questions:**

N/A

**Limitations:**

Please refer to the weaknesses.

**Strengths And Weaknesses:**

Strengths
1. It is the first paper to discuss how Go agents perform against adversarial examples.
2. The paper is in good shape and it is easy to follow.
3. The method is technically sound.

Weaknesses
1. Although this paper is the first one that investigates the robustness of Go agents, I am more concerned about whether the definition of adversarial example in fooling Go agents is different from that in other fields, and this difference determines whether it is meaningful to investigate. As we all know, remarkable characteristics of the adversarial example in the traditional field are the wrong prediction of the model and the meaningless but inconspicuous disturbance itself. I know the authors repeatedly emphasize that using the proposed method will generate meaningless stones and make the agents behave abnormally, but these actions do not conform to the traditional sense of imperceptibility. Moreover, the most surprising point of adversarial examples at present is that there seemingly exists an inevitable phenomenon between the robustness and accuracy of the ML model, that is, adversarial training will reduce the performance of any model in identifying clean samples. But considering the case given in this paper, I tend to believe the lack of appropriate training strategy is the main reason for the misprediction given by the agent, rather than the so-called “adversarial example”.
2. For the evaluation experiments, I am a little confused why these works [1][2][3] cannot be good baselines for evaluating the performance of the proposed method to attack AZ agents.

[1] Zhang, Xuezhou, et al. "Adaptive reward-poisoning attacks against reinforcement learning." ICML, 2020.

[2] Zhang, Huan, et al. "Robust deep reinforcement learning against adversarial perturbations on state observations." NeurIPS, 2020.

[3] Gleave, Adam, et al. "Adversarial policies: Attacking deep reinforcement learning." ICLR, 2020.

---

> ### Author Response · Authors · 2022-08-02
> **Official Comment of Paper5438 Part 2/2**
>
> >  The most surprising point of adversarial examples at present is that there seemingly exists an inevitable phenomenon between the robustness and accuracy of the ML model; that is, adversarial training will reduce the performance of any model in identifying clean samples.
>
> -  As mentioned above, our goal is to find "bugs" for AZ agents, which will shed light on how to improve existing methods. Currently, we do not know whether there's a trade-off between robustness and accuracy in the case of AZ, but we agree with the reviewer that fixing those errors may sacrifice clean performance. The focus of this work is to first discover the adversarial examples in Go agents. We leave the study of the trade-offs as a future work.
>
> > For the evaluation experiments, I am a little confused why these works [1][2][3] cannot be good baselines for evaluating the performance of the proposed method to attack AZ agents.
> - Thank you for pointing out these papers. Since our paper focuses on finding adversarial perturbation of the discrete turn-based game against AZ (MCTS search-based) agents, all of these methods, which are designed on continuous domains without searching, won't work. Additionally, any gradient-based techniques won't work since our target agent policy is search-based, and there is no easy definition of a gradient. Moreover, the purposes of the three papers are quite different from our method, as explained below:
>     - [1] Zhang, Xuezhou, et al. "Adaptive reward-poisoning attacks against reinforcement learning." ICML, 2020.
>         - This paper perturbs a small part of the *rewards* during training to make the agent unable to learn properly. Our work only tries to evaluate the robustness of existing state-of-the-art agents during testing, and we do not perturb the rewards during training.
>     - [2] Zhang, Huan, et al. "Robust deep reinforcement learning against adversarial perturbations on state observations." NeurIPS, 2020.
>         - This paper focuses on adding perturbation to the *observations* of the agent, not the actual states. We perturb the actual state instead of observation, and we focus on discrete-space agents with MCTS, which is not discussed in [2].
>     - [3] Gleave, Adam, et al. "Adversarial policies: Attacking deep reinforcement learning." ICLR, 2020.
>          - This paper tries to fool the opponent by leading the agent to a state far from its training distribution. In comparison, our adversarial is only two steps away from a normal state. We additionally point out three major differences between Gleave, Adam, et al. and our work in our response to [Reviewer 6PMq](https://openreview.net/forum?id=yZ_JlZaOCzv&noteId=7fGJTstolV4).

---

> > ### Comment · Reviewer_nfAu · 2022-08-07
> > **Follow up**
> >
> > I'd really appreciate that the authors give a detailed response to my concerns. However, I am afraid I cannot agree with the NLP examples the authors used to support their opinions. As we all know, the way to generate adversarial examples on the CV, NLP or speech tasks is mainly by applying malignant noises to the raw data. This is because it is almost the only way to generate adversarial examples in the real-world setting, which makes the topic valuable as the hacker always has no access to the intermediate results or states of any deployed DNN model. Back to the scope of this paper, the authors focus on attacking the special intermediate states during playing Go, which is almost impossible for humans to fabricate. So I really do not think it is a practical or meaningful setting for the adversarial community. Moreover, it is still not known whether this so-called "adversarial example" in the paper will decrease the accuracy of Go agents in normal scenarios. As the authors said, lots of players use AZ-like agents to enhance their skills, so they more care about performance of the agents under regular conditions. Sacrificing performance of clean accuracy for a situation that is unlikely to happen makes this issue even more meaningless from the perspective of practicality.
> >
> > Besides, though the authors claim it is the first time to explore adversarial examples in AlphaZero-like Agents, I still think it is the responsibility of the authors to provide a strong and reliable baseline method from previous RL methods because this can more intuitively and accurately evaluate the effectiveness of the method in this paper.
> >
> > Based on these considerations, I still keep my initial score.

---

> > > ### Author Response · Authors · 2022-08-08
> > > **Follow up 2/2**
> > >
> > > > Besides, though the authors claim it is the first time to explore adversarial examples in AlphaZero-like Agents, I still think it is the responsibility of the authors to provide a strong and reliable baseline method from previous RL methods because this can more intuitively and accurately evaluate the effectiveness of the method in this paper.
> > > - As we discussed in the previous comment, none of the baselines can be applied to our setting with discrete states, discrete actions with non-differentiable MCTS procedure in agent policy, and some papers also have quite different focuses (e.g., altering rewards or observations, which is not the same setting as ours). We can certainly list these baselines and report them as inapplicable if the reviewer thinks that is helpful. We feel our method is intuitive enough and can serve as a baseline for future attacks.
> > >
> > > Since our work is the **first demonstrated attacks on Go AI, a setting where adversarial examples were never defined nor demonstrated, we hope the reviewer can understand that many properties for classical adversarial examples are not directly applicable here** (e.g., noises are imperceptible) and many classical thinkings of adversarial examples (e.g., performance vs robustness trade-off) are not always valid here. Our paper studies a novel problem setting proposes non-trivial algorithms and demonstrates surprising results, which we believe that does not belong to the rating of 3: `a paper with technical flaws, weak evaluation, inadequate reproducibility and incompletely addressed ethical considerations.` Hopefully, most of your concerns are addressed now, and we hope you can kindly reevaluate our paper based on our discussions. Thank you.

---

> > > ### Author Response · Authors · 2022-08-08
> > > **Follow up 1/2**
> > >
> > > We thank you for reading our comments carefully and for the constructive reply. However, we still feel there are certain misunderstandings we want to further clarify.
> > >
> > > > This is because it is almost the only way to generate adversarial examples in the real-world setting, which makes the topic valuable as the hacker always has no access to the intermediate results or states of any deployed DNN model.
> > > - We agree with the review that attacking a deployed DNN model without accessing the model (i.e., black-box attack) is an important topic. However, most adversarial attacks start with white-box attacks where the attacker can access the model [1, 2, 3] since it is important to understand the limitation of any model we are using. Our attack is the first attack on Go AI, so we follow the white-box setting.
> > >
> > > [1] Madry, Aleksander, et al. "Towards deep learning models resistant to adversarial attacks." arXiv preprint arXiv:1706.06083 (2017).
> > >
> > > [2] Carlini, Nicholas, and David Wagner. "Towards evaluating the robustness of neural networks." 2017 ieee symposium on security and privacy (sp). Ieee, 2017.
> > >
> > > [3] Ebrahimi, Javid, et al. "Hotflip: White-box adversarial examples for text classification." arXiv preprint arXiv:1712.06751 (2017).
> > >
> > >
> > > > Back to the scope of this paper, the authors focus on attacking the special intermediate states during playing Go, which is almost impossible for humans to fabricate. So I do not think it is a practical or meaningful setting for the adversarial community.
> > >
> > > - First, we argue that although the adversarial examples we found for Go are not "natural" states, when adversarial examples are initially discovered for DNNs, they are not natural, as well - the imperceptible adversarial noises almost never occur in natural data. So **the criticism actually applies to most papers studying adversarial examples**. Moreover, existing works on attacking RL also alter agent states [4,5] in a way that are impossible to happen in real scenarios.
> > >
> > > [5] Huang S, Papernot N, Goodfellow I, Duan Y, Abbeel P (2017) Adversarial
> > > attacks on neural network policies. arXiv preprint arXiv:1702.02284
> > >
> > > [4] Lin, Yen-Chen, et al. "Tactics of adversarial attack on deep reinforcement learning agents." arXiv preprint arXiv:1703.06748 (2017).
> > >
> > > - Additionally, we believe that proving how bad an agent can be even in an unnatural ("almost impossible for humans to fabricate") and adversarial setting is important, and this is the entire focus of the adversarial community. With our found adversarial examples, people can understand that currently, a strong Go AI sometimes still does not have the basic knowledge of Go that an amateur human player can understand, which is really surprising for human Go players (just like how adversarial examples were surprising in image classifiers).
> > >
> > > >  As the authors said, lots of players use AZ-like agents to enhance their skills, so they more care about performance of the agents under regular conditions.
> > >
> > > - In reality, Go players probe the knowledge of AI to learn how AI solves a particular problem by creating fake states. For example, to solve life-and-death problems (i.e., [tsumego](https://senseis.xmp.net/?Tsumego)) with AI, one of the previous works [6] [fill the rest of the board with nonregular stones to make the AI focus on the local battle](https://i.imgur.com/LicjU9I.png). Our results show that these states may mislead the agent, which is not surprising now since we have shown in our paper that the agent is still vulnerable even with easier states (novice player often knows the right moves) and more regular conditions (only two steps away).
> > >
> > >
> > > [6] Shih, Chung-Chin, et al. "A Novel Approach to Solving Goal-Achieving Problems for Board Games." Proceedings of the AAAI Conference on Artificial Intelligence. Vol. 36. No. 9. 2022.
> > >
> > > >  Sacrificing performance of clean accuracy for a situation that is unlikely to happen makes this issue even more meaningless from the perspective of practicality.
> > > - First of all, the purpose of this work is to show the limitations of Go AI, and we do not claim to show any clean performance trade-off in this work, although we empirically found that a more robust agent may have better performance. For example, the DNNs trained by the stronger AZ algorithm are both better and more robust than those that do not.

---

> ### Author Response · Authors · 2022-08-02
> **Official Comment of Paper5438 Part 1/2**
>
> We thank the reviewer for raising the insightful questions that we did not emphasize in the paper. The main concern of the reviewer comes from a classical definition of adversarial example in computer vision. The contribution of our paper becomes immediately clear once this definition is generalized to discrete domains such as NLP and games, as we will discuss in detail below.
>
> > These actions do not conform to the traditional sense of imperceptibility.
> -  We agree that "human imperceptibility" is one important aspect of adversarial examples, but this usually holds only in continuous domains (e.g., vision, speech), not discrete domains (e.g., NLP, board games). In general, given a natural example $s$, we can define an adversarial example $s'$ as another example that is semantically equivalent to $s$ but predicted differently by the model. In continuous domains, "semantically equivalent" can be defined by human imperceptibility. However, in discrete domains, usually, any change can be perceptible. For example, in NLP, given a natural sentence, "This movie had terrible acting," an adversarial example can be "This movie had awful acting," where we change only one word in the sentence to its synonym. The change is **not human imperceptible** (a word has been obviously changed), but humans can easily verify these two sentences should be semantically equivalent, so if the perturbation changes the model's prediction, we know there's an error in the model. There are tens of papers every year studying this kind of word substitution-based adversarial examples in NLP, see [1,2,3,4,5,6] and a benchmarking GitHub repository that collects existing text attacks https://github.com/QData/TextAttack. Our definition of adversarial example is almost identical to this definition in NLP, where we add only 1 or 2 stones and require the perturbed state to be semantically equivalent to the original state (i.e., the added stones do not give any advantage to any players).
>
>     [1] Alzantot, Moustafa Farid et al. "Generating Natural Language Adversarial Examples." EMNLP (2018).
>
>     [2] Jin, Di et al. "Is BERT Really Robust? A Strong Baseline for Natural Language Attack on Text Classification and Entailment." AAAI (2020).
>
>     [3] Ren, Shuhuai et al. "Generating Natural Language Adversarial Examples through Probability Weighted Word Saliency." ACL (2019).
>
>     [4] Li, Linyang et al. "BERT-ATTACK: Adversarial Attack against BERT Using BERT." ArXiv abs/2004.09984 (2020): n. pag.
>
>     [5] Jia, Robin et al. "Certified Robustness to Adversarial Word Substitutions." EMNLP (2019).
>
>     [6] Li, Zongyi et al. "Searching for an Effective Defender: Benchmarking Defense against Adversarial Word Substitution." EMNLP (2021).
>
>
>
> > Why do we want to study adversarial examples for AZ agents?
> - The adversarial examples can be viewed as "bugs" of AZ agents. Finding bugs of machine learning models is important for improving existing models and is one of the main use cases for adversarial examples. For instance, in lines 382--392, our attacker reveals the problem of existing agents that "Agents are sensitive to the ordering of actions in the trajectory", and we propose a simple approach to significantly improve the robustness of the KataGo agent.
> - Additionally, our work highlights that even very strong Go agents like AZ may not be trustworthy, uncovering the potential weakness and limitations of AI. With MCTS, AZ is both more "robust" and "accurate" than humans compared to most ML models in other fields. Many professional players use AZ agents every day to improve their skills, so it is important to fully understand the limitations of these agents.
>
>
> > I tend to believe the lack of appropriate training strategy is the main reason for the misprediction given by the agent, rather than the so-called "adversarial example".
> -  Although we do agree a more "appropriate training strategy" may eliminate some adversarial examples (like adversarial training for classification models), the AZ algorithm we evaluated is currently the state-of-the-art training strategy, and we showcased surprising mistakes it can make and the weakness of the state-of-the-art strategy. We hope our discovery can inspire researchers to propose a more "appropriate training strategy", similar to adversarial training.
> -  We believe that the mistakes we found can be called "adversarial examples" since the agents are correct on the original state but extremely wrong on a perturbed state very similar to the original state; see also our answer to the first question on discussing the definition of "imperceptibility" in discrete space systems.

---

### Official Review · Reviewer_9kDD · 2022-07-08

**Rating:** 5
**Confidence:** 4
**Soundness:** 3 good
**Presentation:** 3 good
**Contribution:** 3 good

**Summary:**

The paper investigates the vulnerability of AlphaZero (AZ) agents against adversarial attacks. The paper first defines an adversarial example for AZ agents as a state that is perturbed by adding a small number of meaningless actions, which can be verified to be semantically similar to the original state. Then, the paper defines an adversarial attack on a state as successful if the policy-value neural network (PV-NN)s produces wrong predicted values, which leads to suboptimal behavior of the agents in the future. The paper proposes an efficient attack method by exploiting the observation that a meaningful action of a state is also a meaningful action of the prior states and reducing the search space for meaningless actions. The paper empirically shows that PV-NNs are vulnerable to adversarial attacks across different AZ agents.

**Questions:**

**Some condition for defining the similarity between two states seems artificial.**

As mentioned in line 182, the authors evaluate the value of s' by $\max(V(s'), 1-V(\mathcal{T}(s', a_s^*)))$. However, it is unclear why we should consider the latter term. If the examiner's prior policy outputs an action different from $a_s^*$ as the optimal action for $s'$ (the authors say the policy "forgets" to consider the best action), then it implies that the examiner, which is much stronger than the target agent, is fooled by $s'$. So, it is natural to expect that $s'$ will also fool the target agent. I think a clearer explanation of why we consider $a_s^*$ to compute the value of $s'$ is needed. Also, it would be better to report the statistics of the difference between the actual value $V(s')$ and the modified value $\max(V(s'), 1-V(\mathcal{T}(s', a_s^*)))$.

Additional question: Does $a_s^*$ denote the action sampled from the prior policy of the target agent or the action sampled after the MCTS?

**The proposed adversarial scenario seems unrealistic.**

The paper generates adversarial examples by adding two meaningless (or pass) moves, but I think this does not occur in reality. For example, a Go player who is playing against an AZ-like agent can not utilize this type of attack to fool the opponent since this attack requires both players to do meaningless moves.



**Limitations:**

The paper clearly states its limitation in Section 5.

**Strengths And Weaknesses:**

Strengths
- To the best of my knowledge, it is the first work that examines the adversarial vulnerability of AZ agents and I think it is novel.
- The paper is well-organized and provides an appropriate figure (Fig. 1) to help readers to understand the new concept of adversarial attacks against AZ agents.
- The experimental design is good and the paper provides very interesting results (e.g., Results on Different Datasets, Robustness of PV-MCTS).

Weaknesses
- Some condition for defining the similarity between two states seems artificial (see Questions below).
- The proposed adversarial scenario seems unrealistic (see Questions below).
- The paper does not provide any quantitative results showing that the proposed attack method degrades the performance of the target agent, though the paper says that the attacked agent will make mistakes way below their level (line 19).
- Some grammatical errors: achieve -> achieves (line 4), indicate -> indicates (line 97), denote -> denotes (line 177), Base -> Based (line 264), .

---

> ### Author Response · Authors · 2022-08-02
> **Official Comment of Paper5438**
>
> > Some condition for defining the similarity between two states seems artificial like $max(V(s'), 1-V(T(s', a^*_s)))$
>
> - The reviewer is correct that when the examiner makes a mistake on state s', the target agent will also be fooled, and this s' will be an adversarial example for both examiner and target agent. However, in this case, our algorithm will think that s, s' are not semantically equivalent and will miss this adversarial example. Therefore, we add the second term $1-V(T(s', a^*_s))$ to make the examiner stronger. Here $a^*_s$ is the examiner's (MCTS) best action of state s, and we use $a^*_s$ as a hint for the examiner. Intuitively, if $s$ and $s'$ are really semantically equivalent, $a^*_s$ will be a good action for s' too. Therefore, using $1-V(T(s', a^*_s))$ will allow MCTS to search from a better initialization and thus get a more accurate solution. Since this is equal to conducting a one-step minimax search, it is not artificial. It can be generalized to a larger minimax tree search when more information of $s$ is hinted.
> - In practice, we found $max(V(s'), 1-V(T(s', a^*_s)))$ is more useful when the target agent is an MCTS agent since the target agent and the examiner are more likely to be fooled together. The attack success rate will drop more than 15% when the simulation count is 50 without using this additional term.
>
> > The proposed adversarial scenario seems unrealistic.
>
> - Like adversarial image attacks, we demonstrate that even AZ agents can be bad at states that are close to nature states. This is may be more surprising than fooling image classifiers since the AZ agents we used in this work already significantly surpass human ability, and it is important to know those agents also made simple mistakes that can be easily verified by humans.
> - We frame this work as finding "bugs" of AZ agents instead of trying to develop a realistic attack to fool them. Finding bugs of machine learning models is important for improving existing models and is one of the main use cases for adversarial examples. For instance, in line 382--392, our attacker reveals the problem of existing agents that "Agents are sensitive to the ordering of actions in the trajectory", and we propose a simple approach to significantly improve the robustness of the KataGo agent.
> -  The main application of AZ-based Go agents is to teach humans the best move under different board states. Even professional players often query the states to the Go agents to learn the best move, in which case the queries may not be natural states. Our work highlights those very strong Go agents like AZ may not be trustworthy for these artificial queries, uncovering the potential weakness and limitations of AI.
>
> > The paper does not provide any quantitative results showing that the proposed attack method degrades the performance of the target agent, though the paper says that the attacked agent will make mistakes way below their level (line 19).
> - A quantitative measure for agent mistakes is defined by the level of the human verifier. When a mistake can be verified by a lower-ranked player, the mistake is more serious and is a lower-level mistake. Note that, although a professional human player will make mistakes, such mistakes can not be verified by amateur human players. Therefore, it is surprising that AZ agents, which are stronger than professional human players, make these low-level mistakes.
>
> >  Grammatical errors:
> -  Thank you for pointing this out. We have fixed the errors in the revision.

---

> > ### Comment · Reviewer_9kDD · 2022-08-08
> > **Response to Paper5438 authors**
> >
> > Thank you for the detailed answers. However, I still have a concern that the adversarial scenario that the authors proposed is not realistic, as the reviewer nfAu also said. So, I decide to keep my score unchanged.

---

> > > ### Author Response · Authors · 2022-08-08
> > > **Follow up**
> > >
> > > Thank you for reading our comments carefully. We still want to clearify about the realistic concern.
> > > > However, I still have a concern that the adversarial scenario that the authors proposed is not realistic.
> > > - In reality, Go players probe the knowledge of AI to learn how AI solves a particular problem by creating fake states. For example, to solve life-and-death problems (i.e., [tsumego](https://senseis.xmp.net/?Tsumego)) with AI, one of the previous works [1] [fill the rest of the board with nonregular stones to make the AI focus on the local battle](https://i.imgur.com/LicjU9I.png). Our results show that these states may mislead the agent, which is not surprising now since we have shown in our paper that the agent is still vulnerable even with easier states (novice player often knows the right moves) and more regular conditions (only two steps away).
> > >
> > > [1] Shih, Chung-Chin, et al. "A Novel Approach to Solving Goal-Achieving Problems for Board Games." Proceedings of the AAAI Conference on Artificial Intelligence. Vol. 36. No. 9. 2022.
> > >
> > > - In addition, we argue that although the adversarial examples we found for Go are not "natural" states, when adversarial examples are initially discovered for DNNs, they are not natural, as well - the imperceptible adversarial noises almost never occur in natural data. Moreover, most existing works [2,3] on attacking RL also alter agent states in a way that are impossible to happen in real scenarios. So this criticism applies to many works on adversarial examples.
> > >
> > > [2] Huang S, Papernot N, Goodfellow I, Duan Y, Abbeel P (2017) Adversarial
> > > attacks on neural network policies. arXiv preprint arXiv:1702.02284
> > >
> > > [3]  Lin, Yen-Chen, et al. "Tactics of adversarial attack on deep reinforcement learning agents." arXiv preprint arXiv:1703.06748 (2017).
> > >
> > > - Finally, we believe that proving how bad an agent can be even in an unrealistic and adversarial setting is important, and this is the entire focus of the adversarial community. With our found adversarial examples, people can understand that currently, a strong Go AI sometimes still does not have the basic knowledge of Go that an amateur human player can understand, which is surprising (just like how adversarial examples were surprising in image classifiers).

---

> > > > ### Comment · Reviewer_9kDD · 2022-08-08
> > > > **Follow up**
> > > >
> > > > Thanks for the reply, but I have to clarify this issue with some questions.
> > > >
> > > > 1.  Who is the adversary and who is the defender in this adversarial threat model?
> > > > 2.  Your proposed adversarial threat model requires both players to make adversarial moves. Should both the players be considered adversaries?

---

> > > > > ### Author Response · Authors · 2022-08-08
> > > > > **Follow up**
> > > > >
> > > > > Thank you so much for willing to further discuss with us!
> > > > >
> > > > > > Who is the adversary and who is the defender in this adversarial threat model?
> > > > >
> > > > > - One way to view our setting is that we have three actors involved: the Go AI, the opponent player, and the attacking adversary. We allow the attacking adversary to slightly perturb one of the states in a given game in a restricted way such that only 1 to 2 stones are added, and these stones are meaningless for the state. Then the Go AI and its opponent player continue to play on the perturbed state starting with the Go AI. The goal of the attacking adversary is to find an example $s'$ so that the Go AI will play an extremely bad move on $s'$ and will lose the game against the opponent. Furthermore, the extremely bad move is so unreasonable such that even human players can easily tell it is incorrect. Hence, we can ignore the opponent and tell that the AI has lost.
> > > > > - The defender is the Go AI agent. Although we are not focusing on defense, we show a strategy to improve the robustness of the Go AI agent based on the vulnerability identified by our attack (line 380).
> > > > >
> > > > >
> > > > > > Your proposed adversarial threat model requires both players to make adversarial moves. Should both the players be considered adversaries?
> > > > >
> > > > > - As we explained above, the adversary is the third person trying to perturb the state, while the Go AI and its opponent continue to play normally. The setting of slightly perturbing system states is valid and has been demonstrated in other RL adversarial attacks, such as [1, 2], where such perturbations are modeled as adversaries in simple Mujoco and Atari environments. The main contribution of our work is to demonstrate that such weakness also exists in the state-of-the-art and superhuman Go AIs in a 2-player game setting with discrete states, actions, and MCTS based policies. We will add this discussion to our paper.
> > > > >
> > > > > [1] Pinto, Lerrel, et al. "Robust adversarial reinforcement learning." International Conference on Machine Learning. PMLR, 2017.
> > > > >
> > > > > [2]  Lin, Yen-Chen, et al. "Tactics of adversarial attack on deep reinforcement learning agents." arXiv preprint arXiv:1703.06748 (2017).
> > > > >
> > > > > - Note that, for most of the AZ agents, we can only input a state to the program with a sequence of legal actions (like the trajectory shown in line 162). Hence, unlike [1, 2] can modify the state directly, the attack adversary can only use legal actions as a perturbation to create adversarial examples. More importantly, all the RL attacks work on continuous space while the state of Go is discrete, so a new attack is needed here.
> > > > >
> > > > >
> > > > > Thank you again for the very insightful questions, which greatly help us to improve our paper. We hope all your concerns have been addressed now, and please kindly let us know if you have any further questions.

---

> > > > > > ### Comment · Reviewer_9kDD · 2022-08-09
> > > > > > **Follow up**
> > > > > >
> > > > > > Thanks for the reply. So, the proposed adversarial scenario introduces the third person (adversary) who intervenes in the middle of the game and adds 1 or 2 stones to the checkerboard. My question is that "should we train AZ agents robust against this type of perturbations that cannot occur in practice?" The authors mentioned that the setting of slightly perturbing system states is valid in the prior work on RL adversarial attacks. However, most of the prior work including [1] focuses on robotic tasks, where modeling uncertainty and variation is natural. For example, when a robot is deployed in the real world, the joint position of the robot can be changed by an extra force such as wind. I suggest that the authors propose a more practical scenario as adversarial literature in other domains.
> > > > > >
> > > > > > [1] Pinto, et al., Robust adversarial reinforcement learning, ICML 2017.

---

> > > > > > > ### Author Response · Authors · 2022-08-09
> > > > > > > **Follow up**
> > > > > > >
> > > > > > > Again, thank you for your insightful question on the practicability of our setting. We give some examples where a Go AI is used with "unnatural" states, and in addition, we argue that the main focus of our paper is to discover the weakness and debug the agent rather than aiming for a practical attack. We discuss in detail below:
> > > > > > >
> > > > > > > > "should we train AZ agents robust against this type of perturbations that cannot occur in practice?"
> > > > > > >
> > > > > > > Besides "playing" starting from the empty board, AZ agents have other applications that will encounter all kinds of legal states. Therefore, making AZ agents more robust is also important.
> > > > > > > - Large-scale MCTS search: When we execute MCTS with millions of simulations, the MCTS will start exploring states that will never be encountered by the agent during playing. If the PV-NN is not general enough to evaluate those states, it might be weaker than the MCTS with fewer simulations. Therefore, if we can improve the PV-NN's ability to generalization on states close to normal states, it can also improve the searching result of MCTS. Our work provides a baseline to evaluate such generalization ability.
> > > > > > > - Proofing: Unlike "playing" which only needs to provide a non-perfect, best-effort attempt to achieve a game's goals, "proofing" needs to provide follow-up strategies for *all* the actions of the opponent, even those looks unreasonable. Proofing is an old but important topic in the Game community. Many algorithms have been proposed, such as alpha-beta search and proof number search. Currently, people are using AZ Agents as a heuristic to make the proofing faster[1]. Furthermore, it is common to introduce some "unnatural" states when proving. For example, to solve life-and-death problems (i.e., [tsumego](https://play.google.com/store/apps/details?id=net.lrstudios.android.tsumego_workshop&hl=en_US&gl=US)) with AI, [1] fill the rest of the board with nonregular stones to make the AI focus on the local battle ([Fig. 10](https://i.imgur.com/FlZ8NNa.png) of [1]).
> > > > > > >
> > > > > > > In general, attack is not the only reason for generating adversarial examples. An important motivation for generating adversarial examples is for debugging --- finding “counterexamples”, defined as the input example that the agent made a wrong prediction, for the agent in order to debug and improve. More specifically, we are trying to find states that the agent makes obvious mistakes, which show the agents are not “perfect” and will shed light on how to improve the current agent. For example, we have shown one example in our paper that the counter-examples obtained by our attack reveal that the AZ agents are vulnerable to the last stone even given the same board, and thus we can try to fix those problems by permuting the sequence before passing into the networks as shown in our paper (Line 380). We believe our developed tool that reveals the “weakness” of AZ agents can be a useful tool for debugging the agents.
> > > > > > >
> > > > > > >
> > > > > > > [1] Shih, Chung-Chin, et al. “A Novel Approach to Solving Goal-Achieving Problems for Board Games.” Proceedings of the AAAI Conference on Artificial Intelligence. Vol. 36. No. 9. 2022.

---

### Official Review · Reviewer_6PMq · 2022-07-10

**Rating:** 6
**Confidence:** 2
**Soundness:** 2 fair
**Presentation:** 2 fair
**Contribution:** 2 fair

**Summary:**

The authors study the robustness of Alphazero-like agents to adversarial attacks in the game of Go and (to a lesser extent) the game of NoGo. The kind of adversarial attack they focus on are 'semantically invariant' or 'meaningless' perturbations, which are moves toward game states that are legal states but that would normally not (significantly) affect either player's win rate. The perturbations are therefore discrete, which rules out standard gradient-based methods for finding adversarial examples. They formally define 'semantically invariance' and use human verifiers to confirm a move's semantic-meaninglessness, since this is usually hard to define. They find such perturbations within a restricted perturbation set (defined using the value function of an 'examiner', which is a strong player that uses deep MCTS search). They further constrain the perturbation set to moves that were not meaningful in past moves during the game and, finally, by confirming whether each remaining move in the candidate perturbation set passes an attack success criterion. The authors verify their method succeeds through several experiments on different agents, datasets, and MCTS depths (0 vs non-zero).

**Questions:**

Lines 190-191: I don't understand why it's easier for a human verifier to verify the 1STEP attack than 2STEP attacks. If it's the case, it isn't explained in a way that's easy to understand.

line 197: The implication here seems to be that not both 1STEP and 2STEP moves are constrained to be meaningless. Based on the introduction, I thought meaningless perturbations were the class that the authors were concerned with producing? The text should be clarified here.

line 245: Why do the authors assume that there are N ≈ 150 (the average number of actions for a state) meaningless moves for each state? Is it not likely to be less than the average number of moves? At very least, perhaps N-1 is more justified?

**Limitations:**

The authors don't extensively discuss the limitations of their work. How easy is it to apply this method to games such as chess or shogi? Are other methods better for those games, which are less combinatorially challenging than Go? What is the use of this method? Is it applicable for adversarial training to improve the robustness of Go-playing agents, or does it have limitations that prevent this?

**Strengths And Weaknesses:**

The paper addresses a somewhat novel problem: How can we efficiently (with limited computation) find meaningless adversarial moves against superhuman opponents in a discrete turn-based game? The authors aptly demonstrate that their method works, not just for Go, but also for a similar game, NoGo, too.

That the moves are found efficiently (which is challenging due to the combinatorial nature of the game) is noteworthy. However, it is unclear how the author's method compares with prior approaches, in particular Gleave et al. (2019). It isn't obvious to me why Gleave et al.'s method would compare disfavourably in terms of computation (though I'm open to an explanation from the authors as to why they think their method is likely an improvement over Gleave et al.'s). This raises a broader point: given the existence of at least one comparable method, the authors ought to have compared their method with it. This said, it's possible that the authors chose not to compare because the compute requirements would be too great; if so, an explanation about this decision should be added to the paper.

The paper is sometimes verbose and is occasionally hard to follow. Some of this is due to occasional grammatical errors (which are rare for the most part). My general advice to the authors would be to make the paper easier to skim. Paragraph headings are one way to achieve this and help to break up dense pages of text, of which this paper has a considerable amount.

I also find myself asking why the authors chose 'semantically invariant/meaningless' moves rather than adversarial moves in general. An explanation about this should be given. Nevertheless, a strength of the paper is that it opens up potential future work such that a network may be adversarially trained to be robust against such attacks.

Since the authors constrain their method to legal board states, they should relate their work to "Natural Adversarial Examples" (Hendrycks et al. 2021).

Overall I expect the significance of the work to be moderate: Go-like games are a very limited set and lessons aren't readily transferable to real world problems (but if they are, then the authors should discuss this). Nevertheless, the paper is well carried out and solves a tough, albeit narrow, problem.

Minor comments:
- Consider changing the title to a question that isn't so trivial a priori. Currently, the title is not descriptive of the actually interesting parts of the paper.
- Please fix the grammar or incomplete sentences in lines 136-138, line 184, line 189, line 194, 264, 566. This is not a complete list; I encourage the authors to look for other errors.
- Fix the sentence: line 193 "Hence, if the actual win rate does not change much, even a much weaker verifier like humans can tell does the extra action benefit a player or not."
- I advise giving a brief formal definition of Q(\cdot) beyond 'action value'.
- Consider hyphenating "turn player" to "turn-player" in line 95
- Line 317: be consistent with Leela capitalization.
- Unless the authors have good reason not to, I weakly recommend making terminology consistent with prior work (Gleave et al. 2019) by replacing 'target agent' with 'victim'. This would also make the term more informative.
- The authors should more clearly state at the start that they do most of their experiments on Go, with minimal experiments on one other game, NoGo.
- The figure would greatly benefit from coloring the perturbation moves. It requires much more effort to understand the figures if one has to look for a tiny letter, whereas it's very easy with color.

---

> ### Author Response · Authors · 2022-08-02
> **Official Comment of Paper5438**
>
> > Question 1: Why it's easier for a human verifier to verify the 1STEP attack than 2STEP attacks.
>
> - In 1STEP attack, the human verifiers only need to check whether the action helps its player get any benefit (such as gaining 1 unit of territory). On the other hand, in 2STEP attack, both actions can help their players get benefits (say one action gains ten territories and another gains nine territories). Therefore, the human verifier needs to be able to tell that the benefits gained by both actions are equal, which is much more complicated (see discussions in Line 192). Hence, we force all the extra actions to be meaningless, so that human verifier only needs to check if both actions get zero territories without the need to compare their values, which is much easier.
>
> > Question 2: line 197: The implication here seems to be that not both 1STEP and 2STEP moves are constrained to be meaningless.
>
> - No, the additional actions of both 1STEP and 2STEP attacks are all meaningless (defined in Line 197). For 1STEP attack, the action will be meaningless once Eq.1 is satisfied. Therefore, we only need to force the actions of 2STEP are meaningless.
>
> > Question 3: line 245: Why do the authors assume that there are N ≈ 150 (the average number of actions for a state) meaningless moves for each state?
>
> - $N$ is the average number of actions for a state, not the meaningless moves for each state. In line 245, we use $\bar N$ to present the number of meaningless moves for each state, which is usually way smaller than 150.
>
> > However, it is unclear how the author's method compares with prior approaches, in particular Gleave et al. (2019).
>
> - Gleave et al. (2019) fooled the opponent by only controlling its agent to play meaningless action. Although conceptually similar, they consider a setting quite different from ours. First, they consider finding actions in a continuous space (e.g., MuJoCo environments), whereas in Go we have a discrete action space. Second, the actions in our setting are in a quite different nature. Compared to the environments Gleave et al. (2019), each step in an intense game such as Go has critical importance, and playing a meaningless action for one player will quickly lead to losing the game. In Gleave et al. (2019), each step is not that important since it might be effective for only 0.1 seconds. Even if a player plays a meaningless action to fool the opponent, it can often be undone in the next time step without leading to a failure of the agent. However, in Go, the player needs to get as much benefit as possible in every step, and it's almost impossible to recover from a meaningless action. Finally, state-of-the-art Go agents have an MCTS component, which is not considered in Gleave et al. (2019). Thus, we cannot directly apply the approach in Gleave et al. (2019) to our setting. We will add these discussions in the revision.
>
> > Why the authors chose 'semantically invariant/meaningless' moves rather than adversarial moves in general
>
> - Our goal is to find adversarial moves to perturb the original state $s$ to $s'$ such that $s$ and $s'$ are semantically equivalent. However, even if $s$ and $s'$ are semantically invariant to the examiner, this might not be obvious to humans. Therefore, we additionally force each action to be meaningless.
>
> > They should relate their work to "Natural Adversarial Examples" (Hendrycks et al. 2021).
>
> - We will cite this paper and discuss it in our revision. This paper discusses a set of natural (semantically preserved) perturbations of image data. In the first half of our paper, we design semantically preserved perturbations of Go states, which shares the same goal with the paper but requires a totally different design due to the discrete space and the game-playing environment.
>
> > How easy is it to apply this method to games such as chess or shogi?
>
> - Like NoGo (end of Section 4), our method can be similarly applied to games like chess and shogi since they satisfy the assumptions we state in Appendix A. To the best of our knowledge, our method is the only one that can find adversarial examples of search-based agents on discrete turn-based games.
>
> > What is the use of this method? Is it applicable for adversarial training to improve the robustness of Go-playing agents?
> - As Go-playing agents are already much stronger than human players, our work is the first demonstrating that even those agents can make mistakes, and such mistakes can be easily verified by humans. Regarding applications, our attack algorithms can be used to find "bugs" in the search-based agents and shed light on debugging the agents. For instance, in lines 382--392, our attacker reveals the problem of existing agents that "Agents are sensitive to the ordering of actions in the trajectory," and we propose a simple approach to significantly improve the robustness of the KataGo agent.
>
> > Other minor comments:
>
> - Thank you for pointing out those issues. We have incorporated them in the revision.

---

### Official Review · Reviewer_W67q · 2022-07-11

**Rating:** 5
**Confidence:** 4
**Soundness:** 3 good
**Presentation:** 2 fair
**Contribution:** 4 excellent

**Summary:**

This work proposed to evaluate the robustness of AlphaZero trained agents for games like Go. The main challenge is that it is hard to define semantically invariant perturbation in discrete games like Go. Therefore, the authors proposed to use action perturbation instead of directly manipulating states to perform attack. The authors constructed two kinds of attack, 1STEP attack and 2STEP attack, and defines what is a qualifying perturbed states given the target agent and a weaker verifier. Realizing the brutal force approach to find an adversarial example is too time-consuming, the authors proposed to make reasonable assumptions that if an action is meaningful for a state, then it is also meaningful for any state before that. Given this assumption, the author proposed a method to reduce the time complexity of finding an adversarial example. The results show that the proposed attack can perform efficient attack on several trained AZ agents under different scenarios.

**Questions:**

1. what is exactly the state space? and how do you calculate the L0 distance between states?
2. intuitively, how to verify that two states s and s' are semantically equivalent? does verifying this require human expertise in the game Go?
3. condition C3 seems quite strong, requiring the verifier to be able to identify a wrong state. How hard is this for a human player to do?
4. In line 235, the authors mentioned "since it only takes on mistake for an agent to lose a game", why is this true? It seems to me if a normal human player makes a small mistake, they would still be able to save the game with some possibility. Also, please define what is a "mistake".
5. Is there a possibility to completely get rid of human participation in training the attacker?

**Limitations:**

The authors mentioned that the limitation of the work is that they are only able to identify the adversarial states but haven't been able to guide the AZ agents to those states. This could be a future work to train an attacker which is also an RL agent to attack the AZ agent.

**Strengths And Weaknesses:**

originality: the paper proposed to solve an interesting problem that is different from existing types of attacks. The proposed solution also seem to work on the mentioned game. This is a strength of the paper.

quality: since the test is done on a challenging AZ game, and the attack success rates are pretty high, I think the paper is of good quality in terms of its experiments and results. However, since this study would involve real humans to verify the results, it can be hard to scale this research, and involving real human to evaluate the results also makes it show to train.

clarity: Some of the description in the paper are kind of confusing for me. First, some mathematical terms are given without any mathematical descriptions. For example, there is no definition of the state space of the game, is that an image or is that a vector containing positions of existing stones of both colors? Second, there are some obscurity in the experiment part, the research involves human participation, however, there is no mentioning of the level of these humans in terms of their expertise level of the game Go, which makes it less convincing. The reason is that not everyone knows the game Go, and even if they know, they may not be experts on this game, and whether an action is meaningless usually also depends on the expertise level of the human.

significance: I believe this result is significant for the AZ model itself as it is different from traditional continuous state space RL attack, and the problem is nontrivial, requiring a lot of computation. On the other hand, an adversarial example for an image is easily understandable for a human, but this is not the case for this game since computers can be better than humans at playing the game. The proposed acceleration reduces the attack computation significantly compared with brute-force approach, which makes a big difference.

---

> ### Author Response · Authors · 2022-08-02
> **Official Comment of Paper5438**
>
> > Clarity 1: "However, since this study would involve real humans to verify the results, it can be hard to scale this research definition of the state space of the game."
>
> - Our method itself doesn't rely on human evaluations. We define our attack in a way that the obtained solution should automatically lead to a human verifiable semantically equivalent example, similar to adversarial attacks in computer vision and NLP. No human evaluation is used for constructing the attack example. The human evaluation is used to verify the quality of the adversarial examples obtained by our algorithm. This kind of human evaluation is also used in other fields, such as NLP attacks, where previous works usually use human evaluation to verify whether the adversarial text created by the attacks is natural and semantically-preserved [1, 2].
>
> [1] Alzantot, Moustafa Farid et al. "Generating Natural Language Adversarial Examples." EMNLP (2018).
>
> [2] Jin, Di et al. "Is BERT Really Robust? A Strong Baseline for Natural Language Attack on Text Classification and Entailment." AAAI (2020).
>
> - Further, since the definition of semantically invariance perturbations mainly involves the value function and MCTS, our attack can be easily generalized to other games, such as NoGo.
>
>
> > Clarity 2: the level of these humans in terms of their expertise level of the game Go
>
> - The level of these humans amateur players are level 2k, 3d, and 5d. (Line 364).
>
> > Question 1: what is exactly the state space? and how do you calculate the L0 distance between states?
>
> - States of Go agents are normally presented by their trajectory (the actions played by both players starting from the empty board) (Line 162). Therefore, the L0 distance is calculated by comparing the edit distance of two trajectories, which will be the number of actions we added (Line 161).
>
>
> > Question 2: intuitively, how to verify that two states s and s' are semantically equivalent? does verifying this require human expertise in the game Go?
>
> - As mentioned in our response to Clarity 1, we will search for an adversarial example without human-in-the-loop. To be specific, we require two states $s$, $s'$ need to have similar win rate (see Eq (1)) and the additional moves (from $s$ to $s'$) are "meaningless". Human evaluation is only used to evaluate whether the adversarial examples obtained by our algorithm are really semantically equivalent to the original state (Line 290, 465).
>
>
> > Question 3: condition C3 seems quite strong, requiring the verifier to be able to identify a wrong state. How hard is this for a human player to do?
>
>  - C3 is one of our goals, where we hope the adversarial examples found by our method can satisfy it. For value attack (Line 215), it is easy since humans can find that $v(s), v(s')$ is different; hence they can tell that at least one of them is wrong. For policy attack (Line 217), we designed several constraints to achieve C3, such as Eq 3, and the best action of $s$ should be the best action of $s'$ (Line 232). Such a constraint does not require human verifiers. We conduct human evaluations just to evaluate whether the final returned state $s'$ of our algorithm is verifiable as we expected. The results (Line 370) show that the human verifiers can verify C3 on most examples pretty easily, like the examples shown in Figure 3. However, there are still examples that we found are hard for the human verifiers, like the examples shown in Figure 4, which require about half an hour of discussion to realize that it is an adversarial example. Usually, with a larger $\eta_{adv}$, the examples will be easier for humans to verify.
>
> > In line 235, the authors mentioned "since it only takes one mistake for an agent to lose a game", why is this true?
>
> - We define an action $a$ as a "mistake" of state $s$ if after playing action $a$, the winrate (according to examiner MCTS) will drop dramatically. When playing against an opponent with a similar level,  making such mistakes will cost the examiner to lose the game.
>
> > Is there a possibility to completely get rid of human participation in training the attacker?
> - We do not need human participation in our attack. See our response to Clarity 1.

---

> > ### Comment · Reviewer_W67q · 2022-08-09
> > **Follow up**
> >
> > Thanks for the reply. Thanks for clarifying that human participation is not required during the training, and it is only used for evaluating the generated adversarial examples. I would really appreciate the author to include some simple introduction of the game nature of Go in, maybe, the supplementary materials since I would expect readers without any background in the game Go will have a hard time understanding the paper. Since this is not similar to image recognition attack, where most people should be able to understand the task nature.

---

> > > ### Author Response · Authors · 2022-08-09
> > > **Follow up**
> > >
> > > Thank you for your suggestion. We agree that without the knowledge of Go, the adversarial example is way less impressive. We will add more background knowledge to the supplementary materials. Note that some information can be found in Appendix E Territory. Thank you again for the help in improving our paper. We hope all your concerns have been addressed now and positively support our paper during the final discussion with AC.

---

### Author Response · Authors · 2022-08-06
**Follow-up questions**

Dear reviewers,

Since the discussion period will end soon (Aug 09 '22 08:00 PM UTC), we are just wondering whether there are any follow-up questions or suggestions about our paper so we can have enough time to respond and improve the draft. Your knowledge and perspective are valuable to us.

Thanks again for your detailed and constructive reviews.

Best,

Paper5438 Authors

---

### Public Comment · ~David_J_Wu1 · 2022-12-10
**Misleading detail in the communication of this paper?**

(Disclosure: I am the primary author of KataGo. And although this openreview is appears long finished, I'm leaving this in the hopes it will still be seen and be helpful to someone.)

Thanks for this interesting work! It was linked to me and brought to my attention some time ago. However, if I understand the method correctly, I would like to raise what I believe is a misleading detail in the communication of the results in this paper, which possibly the authors themselves failed to fully appreciate. While the detail does not invalidate the results, I believe it is likely that the large majority of readers of this paper will not be technically familiar enough to spot the issue, and therefore may come away with a conclusion that, even if it turns out in future work to be true, the evidence in this paper doesn't yet support.

Specifically, this paper gives the impression that AlphaZero agents in Go can be made to return incorrect answers by adding a small number of irrelevant perturbations to the board. However, there are two straightforward ways one could perturb the board:

* A. Apply perturbations to be present already in the stone pattern as of the start of the agent's context move history context window, leaving the recent move history unchanged, i.e. with the agent and opponent's recent behavior unchanged.

* B. Insert additional perturbation moves at the end of the history for all the different AlphaZero-trained nets, as if the perturbations were moves that the agent and the opponent themselves actually just now decided to play.

According to the paper, the authors choose B, but this choice has a much larger semantic effect on the state than choice A. Choice B often asks the neural net to generate a prediction conditional on the extremely strange counterfactual "for the last two moves, both players mutually agreed that two absurd moves were in fact the best moves". This gives a substantially different interpretation to the results than the one most readers may think.

To draw a close analogy:

* Suppose you were shown a Chess tactics puzzle from a real game and asked what move you thought was best, and you solved it correctly.

* A. Then I made one irrelevant change to the corner of board that has no relevance to the tactic and as a result you failed it instead. I can reasonably say I "tricked" you and found a clear adversarial perturbation against you.

* B. Instead suppose I performed that same change by appending two bizarrely weird or bad moves to the game, but promising to you "this tactic is from a game where on the previous turn, two super-GM players played those two moves". Then, if you trusted me, then you might seriously doubt that the obvious tactic you see works, because if it did work, high-level players would never have played those weird moves on the previous turns. And it can't simply be one player blundering - *both* players agreed that the tactic wasn't worth playing and was not worth defending against. So you conclude it must not work, and start looking for weirder options that might make their moves make sense, and you get the puzzle wrong.

In case B, I could still say I tricked you, and it was your failure because the game theoretic value and/or correct move shouldn't depend on what the history was. That is the paper's definition of "semantically equivalent". But while clearly a failure from a game-theoretic perspective, the way in which I tricked you in case B is much less impressive than in case A! And I think most people would agree the semantic equivalence between the original and perturbed situation in B is substantially weaker than that of A. But I think most readers will not think carefully about this because the paper at best only discusses it briefly - indeed most colleagues I've personally talked to who read this paper in fact did come away with the impression that A was what was claimed instead of B, thereby ending up misinformed.

---

> ### Public Comment · ~David_J_Wu1 · 2022-12-10
> **Misleading detail in the communication of this paper? (continued)**
>
> (continued)
>
> There is a section "Agents are sensitive to the ordering of actions in the trajectory" with an accompanying experiment that partly addresses this, but I'm not sure it addresses the fundamental issue. The proposed move order shuffling still leaves the perturbation moves visible in the recent history. Moreover, I'm not sure this section helps the reader understand intuitively how semantically unusual the such perturbations could be, in the same way the Chess puzzle analogy above does.
>
> The choice of B rather than A also may influence other points in the paper. For example, the point that the attack model is important because "such examples may be explored during large-scale MCTS and end up misleading the search". If it often generates states where both players' last moves forced to be directly observed blunders, perhaps B could be especially *unlikely* to generate relevant states. In a minimax tree, states on the optimal line and where only one player blunders are relevant to proving the value of the root, but states reached after *both* players blunder are entirely irrelevant to proving the root value.
>
> However I think the efforts of the authors to do research on this topic is great! There are many known weaknesses in seemingly-superhuman agents in Go that are under-appreciated by the research community, which even affect routine usage of these agents as game analysis tools, and likely many other weaknesses not yet known. I and other on-the-ground engineers have dealt with history dependence before - for example, KataGo already has mitigations against the above issues (which I'm disappointed the authors also did not consider to enable or test!), and Leela Chess Zero devs have also considered this issue in the past. But we have not always had the attention and personal resources to do as thorough experiments as in this paper, so having such work is valuable.
>
> It might be the case that even with A, one could still find a high density of adversarial states - that the impression readers may currently get that a small/irrelevant state change is responsible for the problem instead of history conditioning, is correct! If true, it would have more consequence for future work, since the history dependence is already a deliberate design choice more than a bug (I'm happy to elaborate on how projects like KataGo both made and/or already tried to mitigate this choice). As it stands, the experiments in this paper don't well-distinguish which of these two factors is responsible. I hope the door is open for followup work by the authors or by any other research group, to publish experiments to distinguish these two factors and/or better communication so that more readers can be educated about the results.

---

### Meta-Review · Area_Chair_A82y · 2022-08-26

**Recommendation:** Accept
**Confidence:** Certain

**Metareview:**

This paper proposes a novel challenge setting: adversarial attacks on a discrete observation space sequential decision making problem which has been well studied in recent times — the game of go. While there has been work in recent years on discrete domains such as language, the use of this setting and the notion of "semantic invariance" exploited in lieu of epsilon L-infinity constraints is intuitive and novel. The reviewers by and large quite liked the paper, with the average score primarily being drawn down by one slightly hostile reviewer. Looking over the discussion surrounding this review, I found the authors made a bona fide attempt to address the reviewers concerned, and that the reviewer, while they did respond, did not quite engage with the counter-argument in a way which caused me to further trust their review. I am happy to go with the majority vote here and recommend acceptance.

**Award:**

No

---

### Decision · Program_Chairs · 2022-09-14

Accept